# From Flat Facts to Sharp Hallucinations: Detecting Stubborn Errors via Gradient Sensitivity

**Yee Zhing Liew** [1 2]  **Andrew Huey Ping Tan** [1]  **Anwar P.P. Abdul Majeed** [3 4]

## Abstract

Traditional hallucination detection fails on "Stubborn Hallucinations" — errors where LLMs are confidently wrong. We propose a geometric solution: Embedding-Perturbed Gradient Sensitivity (EPGS). We hypothesize that while robust facts reside in flat minima, stubborn hallucinations sit in sharp minima, supported by brittle memorization. EPGS detects this sharpness by perturbing input embeddings with Gaussian noise and measuring the resulting spike in gradient magnitude. This acts as an efficient proxy for the Hessian spectrum, differentiating stable knowledge from unstable memorization. Our experiments show that EPGS significantly outperforms entropy-based and representation-based baselines, providing a robust signal for detecting high-confidence factual errors.

## 1. Introduction

Large Language Models (LLMs) have catalyzed a paradigm shift in artificial intelligence and enabled remarkable proficiency across generative and reasoning tasks. However, the deployment of these models in critical domains is severely hindered by the phenomenon of **hallucination**. This term refers to the generation of content that is fluent and coherent but factually incorrect (Ji et al., 2023; Zhang et al., 2025). While significant efforts have been directed toward mitigation strategies, the persistence of these errors in state-of-the-art models (Kalai et al., 2025) underscores the urgent need for reliable post-hoc detection mechanisms.

Current approaches to hallucination detection predominantly frame the problem through the lens of predictive uncertainty. Black-box methods operate on the assumption that hallucinations manifest as "confusion" (Farquhar et al., 2024; Tonolini et al., 2024). This state is typically characterized by high entropy or inconsistency across stochastic samples. Similarly, white-box methods often analyze static internal representations to detect deviations in the latent space (Chen et al., 2024; Wang et al., 2025). However, these uncertainty-based paradigms face a critical failure mode known as the **Stubborn Hallucination**. We define Stubborn Hallucinations as factually incorrect predictions generated with high confidence and stability. In these pathological cases, the model has memorized an error or heuristic, resulting in a low-entropy output distribution that mimics the statistical signature of robust factual knowledge. Because the model is "confidently wrong," methods relying on output consistency or probability are rendered ineffective as they fail to discriminate between a robust fact and an entrenched error.

In this work, we diverge from probabilistic heuristics and propose a geometric perspective on hallucination detection. Drawing on generalization theory (Hochreiter & Schmidhuber, 1997; Keskar et al., 2017), we hypothesize that the distinguish-ability between facts and stubborn errors lies not in the output probability, but in the local curvature of the loss landscape. As illustrated in Figure 1 (left), we posit that generalized knowledge resides in flat minima. These are regions supported by redundant features learned from diverse contexts, where the loss remains robust to local parameter perturbations. Conversely, stubborn hallucinations correspond to sharp minima. These arise from the memorization of sparse or noisy patterns (Feldman, 2020). While both regimes may yield low loss and high confidence at the exact minimum, their geometry differs fundamentally. Even a minor perturbation in a sharp basin induces a dramatic spike in the gradient magnitude, whereas a flat basin remains stable.

To operationalize this insight without the prohibitive cost of computing the Hessian, we introduce **Embedding-Perturbed Gradient Sensitivity (EPGS)**. By injecting Gaussian noise into the input embedding space, we actively probe the stability of the optimization landscape. We derive

---

[1]School of Intelligent Manufacturing Ecosystem, Xi'an Jiaotong-Liverpool University, People's Republic of China [2]Department of Computer Science, University of Liverpool, United Kingdom [3]Faculty of Engineering and Technology, Sunway University, Malaysia [4]School of Robotics, Xi'an Jiaotong-Liverpool University, People's Republic of China. Correspondence to: Yee Zhing Liew <yeezhing.liew23@student.xjtlu.edu.cn>.

*Proceedings of the 43rd International Conference on Machine Learning*, Seoul, South Korea. PMLR 306, 2026. Copyright 2026 by the author(s).

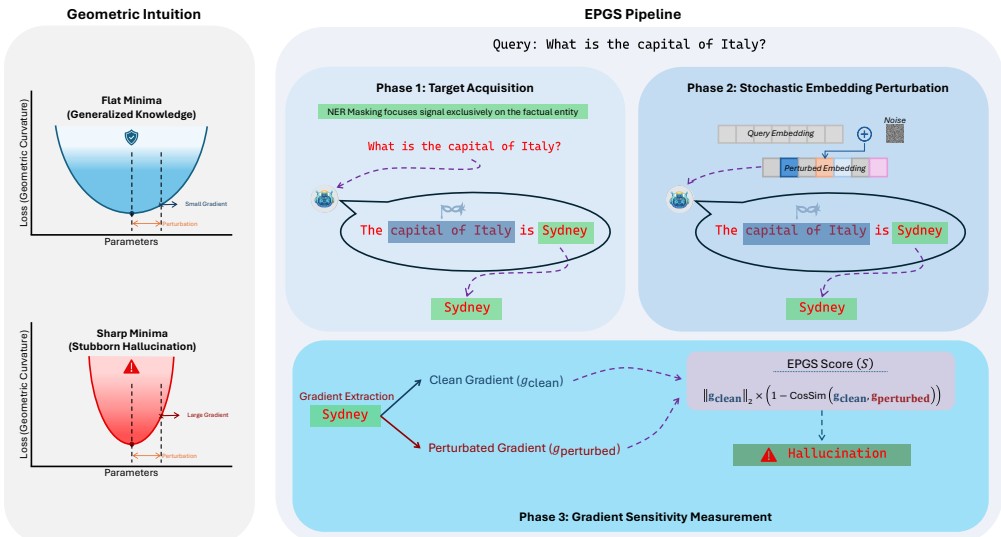

*Figure 1.* **Left:** Geometric intuition underlying our approach. Generalized knowledge resides in flat minima, where a parameter perturbation yields a small gradient. Stubborn hallucinations reside in sharp minima, where the same perturbation induces a large gradient spike. **Right:** The EPGS pipeline consists of three phases. Phase 1 (Target Acquisition) isolates the factual entity using NER masking. Phase 2 (Stochastic Embedding Perturbation) injects Gaussian noise into the query embedding. Phase 3 (Gradient Sensitivity Measurement) computes the EPGS score by comparing the clean and perturbed gradients to detect the geometric instability characteristic of a hallucination.

a theoretical connection showing that input sensitivity serves as a computationally efficient proxy for the Hessian spectrum, effectively measuring the "sharpness" of the model's belief. Our contributions are threefold:

- **Geometric Definition of Hallucination:** We categorize hallucinations based on loss landscape geometry. We distinguish between Transient Hallucinations in unstable regions and Stubborn Hallucinations in sharp minima. This classification provides a theoretical framework explaining why confidence-based metrics fail.

- **EPGS Framework:** We propose a white-box detection method that utilizes gradient sensitivity under embedding perturbation to estimate local curvature. This approach effectively separates robust knowledge from brittle memorization.

- **SOTA Performance:** We demonstrate that EPGS significantly outperforms both entropy-based and representation-based baselines across multiple benchmarks.

## 2. Related Work

### 2.1. Types of Hallucination

Following recent surveys (Zhang et al., 2025; Ji et al., 2023), we categorize hallucinations based on the nature of con-

tradiction. **Input-conflicting hallucinations** encompass generations that deviate significantly from the user's input query or provided source context. These errors are frequently observed in summarization tasks where the model generates details absent from the source document (Huang et al., 2025). A distinct category is **context-conflicting hallucinations**, which refer to self-contradictions within a single generated sequence. In these instances, the generated content remains consistent with the input but becomes logically inconsistent with previously generated tokens, a phenomenon often attributed to "lost-in-the-middle" attention deficits where the model fails to maintain long-range dependencies (Liu et al., 2024).

The most pernicious errors, however, are **fact-conflicting hallucinations**, where the model generates fluent, plausible, but factually incorrect statements that contradict established world knowledge. We further bifurcate this category based on the model's predictive stability. **Transient hallucinations** arise when the model lacks knowledge about a specific entity or concept, resulting in a high-entropy output distribution where stochastic sampling yields diverse, inconsistent answers (Tonolini et al., 2024). In contrast, we define **stubborn hallucination**s as errors where the model consistently converges to the same incorrect fact across multiple stochastic generations. These errors exhibit low entropy and high confidence, mimicking the statistical behavior of robust knowledge (Feldman, 2020). Because these errors are stable, they render traditional consistency-based detection

methods ineffective.

## 2.2. Hallucination Detection

Current approaches to hallucination detection can be broadly classified into three paradigms based on the information source they utilise. The most straightforward approach is direct elicitation, which involve prompting the model to verbalize its confidence or self-evaluate its generation. While intuitive, these methods suffer from severe miscalibration, as LLMs frequently exhibit high verbal confidence in their own (Kadavath et al., 2022; Lin et al., 2022b).

A more robust line of research focuses on output probability consistency (black-box methods). Metrics such as Length-Normalized Entropy (LNE) (Malinin & Gales, 2021) aggregate token probabilities to approximate sequence likelihood. To address the issue of semantic equivalence, advanced techniques like Semantic Entropy (Farquhar et al., 2024) sample multiple trajectories for a single query and measure the entropy over meanings rather than tokens. While effective for detecting transient hallucinations, these methods are inherently limited by their reliance on disagreement. They fail fundamentally when applied to stubborn hallucinations, where the model is confidently wrong and consistently generates the same incorrect meaning, resulting in a false signal of factual validity. To overcome the limitations of surface-level statistics, recent scholarship has turned to internal representation consistency (white-box methods). Approaches such as EigenScore (Chen et al., 2024) and Effective Rank (Wang et al., 2025) analyze the geometry of the hidden state space. These methods posit that factual recall utilizes a higher effective rank and richer feature usage than hallucinated content, which often suffers from representational collapse. However, these methods typically analyze static representations, or "inference at rest" which also fail fundamentally when applied to stubborn hallucinations.

## 3. Geometric View of Hallucinations

### 3.1. Preliminaries and Notation

Let $f_\theta : \mathcal{X} \to \mathcal{Y}$ denote an autoregressive LLM parameterized by $\theta \in \mathbb{R}^d$. Given an input query $x$, the model generates a response sequence $y$. We consider the standard cross-entropy loss $\mathcal{L}(\theta; x, y) = -\sum_t \log P_\theta(y_t \mid y_{<t}, x)$.

We operate in a **reference-free setting** where ground truth is unavailable during inference. Consequently, we treat the model's greedy-decoded response $\hat{y} = \arg\max_y P_\theta(y \mid x)$ as a pseudo-label. Our analysis focuses on the local behavior of the optimization landscape around the converged parameters $\theta^*$ with respect to this fixed prediction $\hat{y}$.

## 3.2. The Geometry of Hallucination

Standard uncertainty quantification methods typically assume that incorrect predictions manifest as high-entropy (flat) output distributions. While this holds for **transient hallucinations** which stem from weakly learned concepts and exhibit low confidence; it fails to account for **stubborn hallucinations**. These are factually incorrect predictions generated with high confidence. Because entropy-based detection relies on distributional flatness, it is inherently blind to these high-confidence errors.

To address this limitation, we reframe the problem from a probabilistic perspective to a geometric one. We propose that while stubborn hallucinations mimic the predictive confidence of true knowledge (zeroth-order similarity), they differ fundamentally in the curvature of their loss landscape (second-order divergence). First, we formalize the stability that characterizes stubbornness:

**Definition 3.1** (Stubborn Hallucination). A prediction $\hat{y}$ is a Stubborn Hallucination if it is factually incorrect ($\hat{y} \neq y_{true}$) yet exhibits $\delta$-stability in the output space against local perturbations. Formally, let $P_{\theta^*}(\cdot \mid x)$ denote the model output at a local minimum $\theta^*$. For a local embedding perturbation $\epsilon \sim \mathcal{U}(B_\rho)$ drawn uniformly from a ball of radius $\rho$, the expected Kullback-Leibler divergence satisfies:

$$\mathbb{E}_\epsilon \left[ D_{KL}\Big( P_{\theta^*}(\cdot \mid x) \,\|\, P_{\theta^*}(\cdot \mid x + \epsilon)\Big) \right] \leq \delta, \quad (1)$$

for a sufficiently small threshold $\delta > 0$.

Crucially, both robust factual knowledge and stubborn hallucinations satisfy this stability condition (low $\delta$). Therefore, zero-order metrics, such as output probability or entropy are insufficient discriminators. To distinguish between them, we hypothesized the second-order properties of the loss landscape as:

*Hypothesis* 3.2 (Curvature Hypothesis). Drawing on generalization theory (Hochreiter & Schmidhuber, 1997; Keskar et al., 2017), we posit that the local geometry of the minimum $\theta^*$ differentiates the nature of the learned information:

**Generalized Knowledge (Facts) → Flat Minima:** Correct predictions supported by redundant features learned from diverse contexts reside in flat minima. Here, the Hessian $\mathbf{H} = \nabla_\theta^2 \mathcal{L}$ has a small spectral norm (small $\lambda_{max}$), indicating robustness to parameter shifts.

**Stubborn Hallucinations → Sharp Minima:** Incorrect but stable predictions arise from overfitting to sparse or noisy patterns (Arpit et al., 2017; Feldman, 2020). These solutions correspond to sharp minima characterized by a large spectral norm, creating narrow basins of attraction.

**Transient Hallucinations → Unstable Regions:** Low-confidence errors reside in regions with non-zero gradient

norms $\|\nabla_\theta \mathcal{L}\| > 0$ or highly anisotropic curvature, representing artifacts of incomplete optimization rather than entrenched memorization.

This distinction is crucial. A Facts is robust because it is supported by a broad consensus of weights updated by diverge gradients. A Stubborn Hallucination is robust only in a narrow sense that it is a memorized singularity. It is "stubborn" because the basin is deep (high likelihood), but "sharp" because it lacks the support of neighbouring parameters.

To empirically validate this geometric distinction, we explicitly computed the top Hessian eigenvalue ($\lambda_{\max}$) via power iteration. As detailed in Appendix C.1, our analysis confirms a strong correlation ($r = 0.855$) between the EPGS score and true Hessian sharpness, with stubborn hallucinations residing in significantly sharper basins (average $\lambda_{\max}$ is $2.5\times$ larger) compared to robust facts.

### 3.3. Input-Parameter Isomorphism

While Hypothesis 3.2 offers a geometric criterion for distinguishing stubborn hallucinations from generalized knowledge, explicitly computing the full Hessian spectrum for LLMs is computationally prohibitive. To circumvent this bottleneck, we introduce **Embedding-Perturbed Gradient Sensitivity (EPGS)**. This method exploits a local functional relationship between the input embedding space and the parameter space. While the mapping from input to output in a deep Transformer is highly non-linear, we posit that locally, a perturbation in the embedding space induces a specific trajectory of activation shifts that can be mapped to an equivalent perturbation in the parameter space of the final layer.

**Lemma 3.3** (First-order Gradient Sensitivity Approximation). *Let $f_\theta$ be a deep neural network decomposed into a body $\Phi$ and a final block $\mathcal{T}_{last}$. For a small input embedding perturbation $\delta$, there exists an induced parameter perturbation $\nu_\delta$ in the final block such that the shift in the loss landscape is locally invariant:*

$$\mathcal{L}(\theta^*, \mathbf{E} + \delta, \hat{y}) \approx \mathcal{L}(\theta^* + \nu_\delta, \mathbf{E}, \hat{y}). \qquad (2)$$

*This equivalence holds up to first-order Taylor approximation around the current activation point and is not claimed to be exact beyond local neighborhoods.*

*Proof.* We utilize the Input-Output Jacobian of the network body to map embedding noise to hidden state noise. We then construct a rank-1 update to the final block's weights that produces an identical shift in the pre-activation of the output. See Appendix A.3 for the complete derivation involving deep non-linear architectures. $\qquad \square$

This equivalence implies that model stability with respect to

input perturbations acts as a high-fidelity proxy for parameter stability. We now formalize the relationship between the gradient norm under input noise and the Hessian spectral properties.

**Theorem 3.4** (Gradient Sensitivity Bounds Hessian Curvature). *Assume $\theta^*$ is a local minimum where $\nabla_\theta \mathcal{L}(\theta^*; x, \hat{y}) = 0$. Under the perturbation equivalence established in Lemma 3.3, the gradient norm with respect to the parameters at the perturbed input is bounded by the spectral radius of the Hessian $\mathbf{H}$:*

$$\|\nabla_\theta \mathcal{L}(\theta^*; x + \epsilon, \hat{y})\|_2 \lesssim \lambda_{\max}(\mathbf{H}) \cdot \|\nu_\epsilon\|_2. \qquad (3)$$

*Proof.* We perform a second-order Taylor expansion of the loss around $\theta^*$. Given that the first-order gradient vanishes at the local minimum, the leading non-zero term is determined by the Hessian $\mathbf{H}$. The gradient at the perturbed state can be approximated as $\nabla_\theta \mathcal{L}(\theta^* + \nu_\epsilon) \approx \nabla_\theta \mathcal{L}(\theta^*) + \mathbf{H}\nu_\epsilon = \mathbf{H}\nu_\epsilon$. Consequently, the magnitude of the gradient vector is bounded by the operator norm of the Hessian, which corresponds to its largest eigenvalue $\lambda_{\max}$. (See Appendix A.4 for the full proof). $\qquad \square$

Theorem 3.4 provides the rigorous justification for utilizing input sensitivity as a curvature metric. It establishes a direct link between the magnitude of the gradient under input noise and the sharpness of the minimum:

**If $\hat{y}$ is a Stubborn Hallucination (Sharp Minimum):** $\lambda_{\max}(\mathbf{H})$ is large. Even minor input noise $\epsilon$ translates to a perturbation $\nu_\epsilon$ that forces the model up the steep walls of the basin, resulting in a distinct gradient spike. **If $\hat{y}$ is a Transient Hallucination (Unstable Region):** The model resides in a non-convex or non-converged region (e.g., a saddle point) characterized by high local variance. Here, the gradient direction is highly volatile under perturbation. As we define in the subsequent formulation, the EPGS score explicitly accounts for this directional instability, ensuring that transient hallucinations are detected alongside stubborn ones. **If $\hat{y}$ is a Fact (Flat Minimum):** $\lambda_{\max}(\mathbf{H})$ is small. The loss basin is wide, ensuring that the gradient induced by perturbation remains negligible ($\approx 0$), indicating robust knowledge retention.

## 4. Methodology

As illustrated in Figure 1 (Right), our proposed method **Embedding-Perturbed Gradient Sensitivity (EPGS)**, is structured into three distinct phases: Target Acquisition, Embedding Perturbation, and Sensitivity Measurement.

### 4.1. Phase 1: Target Acquisition and Entity Masking

To accurately measure the model's certainty regarding a specific fact, we must isolate the gradient signal corresponding

to the core entity, filtering out noise from syntactic filler words.

**Pseudo-Label Generation** We input the clean query $x$ (e.g., *"Who directed Inception?"*) into the LLM and obtain the greedy decoded answer $\hat{y}$ (e.g., *"Inception was directed by Christopher Nolan"*). Since we operate in a reference-free setting, this response $\hat{y}$ serves as the pseudo-label for subsequent analysis.

**Entity Extraction** To focus the gradient analysis solely on factual content, we employ an external Named Entity Recognition (NER) pipeline (`BERT-base-NER`) to identify the primary entity within $\hat{y}$ (e.g., *"Christopher Nolan"*). Let this entity sequence be denoted as $y_{entity}$.

**Gradient Masking** We construct a binary target mask $\mathbf{M}$ that isolates the tokens corresponding to $y_{entity}$. When computing the cross-entropy loss, all non-entity tokens are assigned an ignore index. Consequently, the backpropagated gradients are derived exclusively from the model's ability to predict the key entity, effectively masking out structural variations such as *"was directed by."*

## 4.2. Phase 2: Stochastic Embedding Perturbation

To rigorously approximate the local curvature described in Lemma 3.3, we inject noise directly into the continuous embedding space rather than relying on discrete prompt engineering or paraphrase. Let $\mathbf{E} \in \mathbb{R}^{L \times d}$ represent the dense embedding matrix of the input sequence. We introduce a stochastic perturbation $\delta$ drawn from an isotropic Gaussian distribution:

$$\mathbf{E}_{perturbed} = \mathbf{E} + \delta, \quad \text{where } \delta \sim \mathcal{N}(0, \sigma^2 \mathbf{I}). \quad (4)$$

This continuous perturbation $\mathbf{E}_{perturbed}$ is passed forward through the transformer, allowing us to explore the immediate neighborhood of the loss basin with high granularity. This serves as a computationally efficient proxy for Hessian-based sharpness without requiring second-order optimization.

## 4.3. Phase 3: Gradient Sensitivity Measurement

In the final phase, we quantify the instability of the prediction. We compute the gradients with respect to the last transformer block parameters $\theta_{last}$ for both the clean input ($g_{clean}$) and the perturbed input ($g_{perturbed}$):

$$g_{clean} = \nabla_{\theta_{last}} \mathcal{L}_{masked}(f_\theta(\mathbf{E}), \hat{y}) \quad (5)$$

$$g_{perturbed} = \nabla_{\theta_{last}} \mathcal{L}_{masked}(f_\theta(\mathbf{E}_{perturbed}), \hat{y}) \quad (6)$$

**EPGS Score** While Theorem 3.4 utilizes the gradient *norm* to bound the Hessian spectrum, our practical metric relies on directional divergence to ensure robustness

against gradient scaling issues. The connection is geometric: Theorem 3.4 establishes that in sharp minima (stubborn hallucinations), the high spectral radius $\lambda_{\max}$ induces a large gradient displacement vector $\Delta g = g_{perturbed} - g_{clean}$. In high-dimensional parameter spaces, a large random displacement vector $\Delta g$ is statistically likely to be orthogonal to the original gradient $g_{clean}$, resulting in significant angular deviation. Consequently, we define the EPGS score by explicitly coupling the gradient's magnitude with its directional volatility:

$$\mathcal{S} = \underbrace{\|g_{\text{clean}}\|_2}_{\substack{\text{Local Geometry} \\ \text{(Curvature Scale)}}} \cdot \underbrace{(1 - \text{CosSim}(g_{\text{clean}}, g_{\text{perturbed}}))}_{\substack{\text{Stochastic Instability} \\ \text{(Directional Divergence)}}} \quad (7)$$

where $\text{CosSim}(u, v) = \frac{u \cdot v}{\max(\|u\|_2, \epsilon) \cdot \max(\|v\|_2, \epsilon)}$ with a stabilization constant $\epsilon = 1e^{-8}$ to prevent division by zero.

This decomposition is critical for robustness. The **magnitude term** captures the local geometry (steepness) of the loss landscape, acting as a scaling factor that reflects the model's baseline "struggle" to maintain the prediction. The **directional term** captures the instability, measuring how easily the optimization path is disrupted by noise. This dual formulation allows EPGS to detect diverse hallucination types:

- **Stubborn Hallucinations (Sharp Minima):** High $\|g_{\text{clean}}\|_2$ due to steepness + Low CosSim $\rightarrow$ **High** $\mathcal{S}$

- **Transient Hallucinations (Unstable Regions):** High $\|g_{\text{clean}}\|_2$ + Low CosSim $\rightarrow$ **Maximal** $\mathcal{S}$

- **Robust Facts (Flat Minima):** Low $\|g_{\text{clean}}\|_2$ + High CosSim $\approx 1 \rightarrow$ **Low** $\mathcal{S} \approx 0$

By combining magnitude and direction, EPGS provides a unified metric that is sensitive to both the sharpness of memorized errors and the instability of confused generation.

## 5. Experiments

We validate our method on two distinct testing scenarios to ensure both broad applicability and specific robustness against high-confidence errors (stubborn hallucinations).

### 5.1. Experiment Setup

**Dataset Selection.** We select a diverse suite of datasets to rigorously evaluate hallucination detection across the spectrum of factual recall and logical reasoning. To assess Open-Domain Knowledge and Factuality, we rely on three distinct benchmarks. First, we utilize **TriviaQA** (Joshi et al.,

2017), a large-scale reading comprehension dataset derived from trivia enthusiasts. Complementing this is **SQuAD** (Rajpurkar et al., 2016), which focuses on reading comprehension based on Wikipedia articles to evaluate the model's ability to ground answers in provided context. We further include **Natural Questions (NQ)** (Kwiatkowski et al., 2019), comprising real user queries from Google search. Beyond factuality, we evaluate Reasoning Robustness using **Simple Variations on Arithmetic Math Word Problem (SVAMP)** (Patel et al., 2021). Unlike standard math datasets, SVAMP applies structural variations to elementary word problems to probe for superficial heuristic matching.

**Model Selection.** We evaluate our methodology on three diverse open-weight LLMs. We utilize **Llama-2-7b** (Touvron et al., 2023), **Llama-3-8b** (Grattafiori et al., 2024), **Mistral-7b-v0.1** (Jiang et al., 2023).

**Baselines.** We benchmark our proposed method against recent hallucination detection metrics, categorized into black-box and white-box approaches. For Black-box methods, which rely solely on output probabilities and generated text, we evaluate: (1) **Length-normalized Entropy (LN-Entropy)** (Malinin & Gales, 2021) (2) **Semantic Entropy (SE)** (3) **Discrete Semantic Entropy (DSE)** (Farquhar et al., 2024) and (4) **P(False)** (Kadavath et al., 2022). For White-box methods, which leverage internal model representations similar to our approach, we compare against: (5) **Eigenscore** (Chen et al., 2024) and (6) **Effective Rank** (Wang et al., 2025).

**Labelling Strategy.** To formulate hallucination detection as a binary classification task, we derive ground truth labels by comparing the model's anchor generation $\hat{y}$ against the reference answer $y_{\text{ref}}$. We utilize two complementary metrics to ensure robust evaluation: (1) **BERTScore** (Zhang* et al., 2020), which measure semantic similarity; and (2) **SQuAD-F1** (Rajpurkar et al., 2016), which measures lexical overlap.

**Evaluation Metric** We report the Area Under the Receiver Operating Characteristic curve (AUROC) (Bradley, 1997). This threshold-independent metric evaluates how effectively the EPGS Score $\mathcal{S}$ separates hallucinations from correct answers.

**Implementation Details** All experiments were conducted on a single NVIDIA RTX 4090 GPU (24GB VRAM) running Python 3.11 and PyTorch 2.5. For the generation of pseudo-response, we employ a low temperature setting ($T = 0.1$) to minimize stochasticity and focus on the model's most likely reasoning path. In the perturbation phase, we inject Gaussian noise with a magnitude of $\epsilon = 0.1$ into the embeddings. To compute the EPGS score,

we extract the gradients with respect to the parameters of the entire final Transformer block.

## 5.2. Result of General Hallucinations Detection

In this experiment, we evaluate performance across the full validation sets of the datasets. This setting assesses the method's baseline capability to distinguish correct answers from incorrect ones in a standard inference pipeline, encompassing the full range of uncertainty profiles from confused, high-entropy outputs to plausible hallucinations.

**Main Results and Analysis.** Table 1 details the hallucination detection performance across three backbone architectures and four datasets. Our proposed method EPGS consistently securing the highest AUROC scores across all 12 model-dataset configurations. The results support three primary conclusions regarding the geometric nature of hallucinations.

**Active Curvature Probing vs. Static Representation (vs. White-Box Methods)** A critical distinction between EPGS and existing white-box methods lies in the transition from static analysis to dynamic probing. Baselines such as *Eigenscore* (Chen et al., 2024) and *Effective Rank* (Wang et al., 2025) assess the model's uncertainty by analyzing the covariance or rank of the internal hidden states at rest. While these metrics capture representational collapse, they are blind to the local geometry of the loss landscape. In contrast, EPGS operationalizes Theorem 3.4, using embedding perturbations to actively approximate the Hessian spectrum. The substantial performance gap observed, for instance, on TriviaQA with Llama-3-8B, where EPGS surpasses Eigenscore by over 9%, validates the *Curvature Hypothesis* (Hypothesis 3.2). It suggests that stubborn hallucinations are not necessarily characterized by degenerate internal states (which static methods detect), but rather by their residence in sharp minima. EPGS effectively isolates these errors by detecting the spike in gradient magnitude ($\|\nabla_\theta \mathcal{L}\|$) induced by the steep walls of the error basin, a signal inaccessible to static representation analysis.

**Geometric Sharpness vs. Predictive Confidence (vs. Black-Box Methods)** Standard uncertainty quantification methods, such as *Length-Normalized Entropy* (Malinin & Gales, 2021) and *Semantic Entropy* (Farquhar et al., 2024), rely on the assumption that hallucinations manifest as low-confidence (high-entropy) generations. This assumption fails for Stubborn Hallucinations, where the model is confidently wrong. The failure of entropy-based methods is most evident in knowledge-intensive tasks; for example, on SQuAD (Llama-2-7B), Semantic Entropy achieves only 0.4839 AUROC, indicative of random guessing. Conversely, EPGS achieves 0.6924, confirming that while the output

*Table 1.* Comparison of **General Hallucination Detection** methods across Llama-2-7B, Llama-3-8B, and Mistral-7B-v0.1 using `BERTScore` for labeling. We compare Black Box methods—Length-normalized Entropy (LNE), $P$(False) (PF), Semantic Entropy (SE), and Discrete SE (DSE)—and White Box methods—Eigenscore (ES) and Effective Rank (ER)—against our proposed method, EPGS. All values reported are AUROC scores (higher is better). The best results are highlighted in **bold** and the second-best results are underlined.

| Model | Dataset | Black Box | | | | White Box | | |
|---|---|---|---|---|---|---|---|---|
| | | LNE | PF | SE | DSE | ER | ES | **EPGS (Ours)** |
| Llama-2-7B | TriviaQA | 0.6079 | 0.6312 | 0.7080 | 0.7122 | 0.7308 | 0.7224 | **0.7629** |
| | SQuAD | 0.6318 | 0.6620 | 0.4839 | 0.6654 | 0.6822 | 0.6812 | **0.6924** |
| | NQ | 0.6721 | 0.5828 | 0.6869 | 0.6897 | 0.6778 | 0.6782 | **0.6951** |
| | SVAMP | 0.6464 | 0.7085 | 0.7835 | 0.7801 | 0.7860 | 0.7808 | **0.8947** |
| Llama-3-8B | TriviaQA | 0.5976 | 0.6755 | 0.7082 | 0.7092 | 0.7174 | 0.7226 | **0.8127** |
| | SQuAD | 0.6232 | 0.5279 | 0.6682 | 0.6686 | 0.6740 | 0.6775 | **0.7742** |
| | NQ | 0.6369 | 0.6642 | 0.6875 | 0.6870 | 0.6947 | 0.6972 | **0.7705** |
| | SVAMP | 0.7612 | 0.6033 | 0.9004 | 0.9145 | 0.9361 | 0.9371 | **0.9732** |
| Mistral-7B-v0.1 | TriviaQA | 0.6214 | 0.7318 | 0.7701 | 0.7689 | 0.7730 | 0.7760 | **0.8289** |
| | SQuAD | 0.6442 | 0.5459 | 0.6905 | 0.7013 | 0.6883 | 0.6826 | **0.7668** |
| | NQ | 0.6474 | 0.6629 | 0.7214 | 0.7256 | 0.7139 | 0.7128 | **0.7712** |
| | SVAMP | 0.7129 | 0.7684 | 0.8747 | 0.8773 | 0.9002 | 0.9037 | **0.9337** |

distribution is flat (confident), the underlying optimization landscape is sharp. By probing the second-order properties of the loss landscape rather than zeroth-order output probabilities, EPGS successfully exposes the brittleness of high-confidence memorization.

**Sensitivity to Transient Instability in Reasoning** The method demonstrates exceptional efficacy on reasoning-heavy tasks, achieving near-perfect separation on SVAMP (e.g., 0.9732 AUROC on Llama-3). Unlike factual recall, where errors often stem from memorized singularities (sharp minima), mathematical reasoning errors frequently correspond to Transient Hallucinations, non-converged, unstable regions of the loss landscape. In these regions, the gradient is not only large but directionally volatile. The dual formulation of the EPGS score (Eq. 7) is particularly advantageous here. The directional term $(1 - \text{CosSim})$ captures the stochastic instability of the optimization path, while the magnitude term captures the lack of convergence. The consistent improvement across architectures (from Llama-2 to Mistral-7B) indicates that this geometric distinctiveness between valid reasoning (flat/stable) and heuristic failure (unstable) is a fundamental property of LLMs that EPGS robustly exploits.

### 5.3. Result of Stubborn Hallucinations Detection

We evaluate the method against **Stubborn Hallucinations** where "confidently wrong" errors pose a fundamental challenge to black-box consistency methods. To isolate these pathological cases, we construct a *Stubbornness Subset* by generating $k = 5$ stochastic responses for each query with temperature $T = 1.0$. We filter the dataset to retain only those instances where the model exhibits high semantic consistency, effectively removing "confused" generations.

This subset comprises both Generalized Knowledges (consistently correct) and Stubborn Hallucinations (consistently incorrect).

Crucially, because output entropy is minimized for both categories in this subset, they are indistinguishable to metrics relying on predictive stability. This scenario serves as a critical stress test, isolating the method's ability to leverage gradient curvature to discriminate between grounded knowledge and memorized errors. Table 2 presents the performance comparison.

**Collapse of Uncertainty Metrics** The results in Table 2 confirm that standard uncertainty quantification breaks down in the face of stubborn hallucinations. On the Llama-2-7B SQuAD benchmark, SE which relies on disagreement among generations, degrades to an AUROC of 0.5842, performing only marginally better than random guessing. Similarly, confidence-based methods like PF fail to generalize, dropping to 0.4978 on Llama-3-8B SQuAD. This empirical failure validates our premise: when a model has memorized an incorrect fact, it collapses into a low-entropy state that mimics true knowledge, rendering zero-order (probability) and first-order (entropy) metrics ineffective.

**Curvature as the Discriminator** In contrast, EPGS maintains robust detection performance, achieving the highest AUROC across all benchmarks. On the challenging Llama-2 SQuAD split, EPGS achieves 0.7373, outperforming Semantic Entropy by over 15% and the strongest white-box baseline (Eigenscore) by nearly 14%. This performance gap highlights the orthogonality of predictive confidence and geometric curvature. While stubborn hallucinations share the high confidence of facts (flat probability distributions), EPGS reveals that they reside in topologically distinct re-

*Table 2.* Performance comparison on the **Stubborn Hallucination Subset** across Llama-2-7B and Llama-3-8B using `BERTScore` for labelling. We evaluate Black Box methods (LNE, PF, SE, DSE) and White Box methods (ER, ES) against our proposed method (EPGS). All values are AUROC scores. The best results are highlighted in **bold** and the second-best results are underlined.

| Model | Dataset | Black Box | | | | White Box | | |
|-------|---------|------|------|------|------|------|------|------------|
| | | LNE | PF | SE | DSE | ER | ES | **EPGS (Ours)** |
| Llama2-7B | TriviaQA | 0.6114 | 0.5372 | 0.6873 | 0.6525 | 0.6803 | 0.6708 | **0.6976** |
| | SQuAD | 0.6222 | 0.5796 | 0.5842 | 0.5814 | 0.5926 | 0.6003 | **0.7373** |
| | NQ | 0.6097 | 0.6753 | 0.6116 | 0.6022 | 0.5819 | 0.5778 | **0.7023** |
| Llama3-8B | TriviaQA | 0.5818 | 0.6262 | 0.6631 | 0.6945 | 0.6867 | 0.6692 | **0.7187** |
| | SQuAD | 0.6637 | 0.4978 | 0.5918 | 0.5803 | 0.6094 | 0.5857 | **0.7816** |
| | NQ | 0.5360 | 0.6547 | 0.7149 | 0.7606 | 0.7565 | 0.7565 | **0.7656** |

gions of the loss landscape as stated in Hypothesis 3.2. The superior performance of EPGS confirms that input-gradient sensitivity serves as a high-fidelity proxy for this Hessian sharpness, successfully recovering the signal that static representation methods and entropy methods miss.

### 5.4. Ablation Studies

**Gradient Source Location** Our ablation study in Table 3 identifies the optimal source for gradient extraction, demonstrating that the Last Transformer Block consistently maximizes detection performance across all architectures. We observe a clear hierarchy: while the Middle Transformer Block captures partial semantic features, it lacks the final consolidation of the model's belief state, resulting in performance drops. Conversely, the LLM Head and Final Layer Norm exhibit severe degradation, often approaching random performance on knowledge-intensive tasks. This failure at the output stage suggests that the projection layer's gradients are masked by probability saturation in high-confidence hallucinations; in contrast, the dense parameters of the final transformer block retain the latent geometric "sharpness" necessary to distinguish robust knowledge from brittle memorization.

**Perturbation Magnitude Sensitivity** Table 4 presents the ablation study on the Gaussian noise magnitude $\epsilon$, demonstrating that EPGS exhibits remarkable stability across orders of magnitude ($\epsilon \in [0.001, 1]$). The performance variance remains minimal (less than 2% fluctuation), confirming that the geometric distinction between robust facts and stubborn hallucinations is a fundamental topological property rather than an artifact of specific hyperparameter tuning. We observe a subtle domain-specific trend: smaller perturbations ($\epsilon = 0.001$) marginally favor TriviaQA, likely providing the high-resolution probing necessary to detect the extremely narrow basins of rote memorization, whereas larger perturbations ($\epsilon = 1$) are well-tolerated in SQuAD and NQ. This resilience implies that the "flat" basins of generalized knowledge are sufficiently wide to accommodate significant embedding shifts without inducing gradient

spikes, validating the method's practical reliability without the need for extensive per-task sweeping.

## 6. Limitations

While EPGS provides a principled geometric signal for hallucination detection, several limitations exist.

**White-box access** EPGS relies on direct access to model gradients ($\nabla_\theta \mathcal{L}$), restricting its application to open-weight architectures.

**Computational latency** Although more efficient than Hessian computation, EPGS requires backward passes, which are inherently more expensive than standard inference. This introduces a latency overhead (Table 8) that may constrain deployment in real-time or resource-limited environments compared to purely sampling-based methods.

**Binary classification** The current framework produces a scalar score for binary detection. While our theory distinguishes between transient (unstable) and stubborn (sharp) errors, the metric does not yet explicitly output these fine-grained categories as diagnostic labels.

**Sensitivity to rare facts** Long-tail factual knowledge often relies on rote memorization, resulting in sharp loss basins similar to stubborn errors (Feldman, 2020). This may lead EPGS to flag rare facts as hallucinations (false positives). However, we argue this conservative behavior is desirable in high-stakes settings, as such "brittle" knowledge lacks the geometric robustness of generalized concepts.

**Entity-Centric** Our current framework relies on an entity extraction mechanism (Phase 1) to isolate the gradient signal and prevent syntactic dilution. Consequently, EPGS is inherently entity-centric, making it highly effective for factoid-based question answering like TriviaQA, SQuAD. However, this reliance poses a limitation when evaluating complex, abstract reasoning tasks, open-ended creative writing, or code generation, where a hallucination may span an

*Table 3.* Ablation study on the gradient extraction location. We compare extracting gradients from the Middle Transformer Block (Mid Trans.), the Last Transformer Block (Last Trans.), the Final Layer Norm (Layer Norm), and the LLM Head. The best performance (AUROC) for each dataset is highlighted in **bold**.

| Location | Llama-2-7B | | | | Llama-3-8B | | | | Mistral-7B | | | |
|---|---|---|---|---|---|---|---|---|---|---|---|---|
| | Trivia | SQuAD | NQ | SVAMP | Trivia | SQuAD | NQ | SVAMP | Trivia | SQuAD | NQ | SVAMP |
| Mid Trans. | 0.7546 | **0.6965** | 0.6484 | 0.8031 | 0.7980 | 0.6866 | 0.6822 | 0.9318 | **0.8303** | 0.6791 | 0.6834 | 0.8170 |
| Last Trans. | **0.7629** | 0.6924 | **0.6951** | **0.8947** | **0.8127** | **0.7742** | **0.7705** | **0.9732** | 0.8289 | **0.7668** | **0.7712** | **0.9337** |
| Layer Norm | 0.6030 | 0.5069 | 0.4604 | 0.5372 | 0.7360 | 0.5928 | 0.6005 | 0.7628 | 0.7636 | 0.5182 | 0.6784 | 0.6916 |
| LLM Head | 0.5848 | 0.5099 | 0.4928 | 0.6378 | 0.7450 | 0.5474 | 0.5964 | 0.7577 | 0.7614 | 0.4594 | 0.6350 | 0.6763 |

*Table 4.* Ablation study on the magnitude of Gaussian noise ($\epsilon$) added during gradient sensitivity analysis. We compare performance across different noise levels $\epsilon \in \{0.001, 0.01, 0.1, 1\}$ for Llama-2-7b and Llama-3-8b. The best result for each column is highlighted in **bold**.

| $\epsilon$ | Llama-2 | | | Llama-3 | | |
|---|---|---|---|---|---|---|
| | Trivia | SQuAD | NQ | Trivia | SQuAD | NQ |
| 0.001 | **0.7748** | 0.7847 | 0.7568 | **0.8130** | **0.7960** | 0.7818 |
| 0.01 | 0.7640 | 0.7381 | 0.7461 | 0.7972 | 0.7932 | 0.7830 |
| 0.1 | 0.7629 | 0.6924 | 0.6951 | 0.8127 | 0.7742 | 0.7705 |
| 1 | 0.7561 | **0.7979** | **0.7792** | 0.8067 | 0.7945 | **0.7985** |

entire logical concept rather than a discrete named entity. Extending geometric curvature probing to these entity-less, free-form generations remains an open challenge for future work.

## 7. Conclusion

In this work, we address the critical challenge of Stubborn Hallucinations, where LLMs generate factually incorrect content with high confidence and stability. We propose **Embedding-Perturbed Gradient Sensitivity (EPGS)**, a geometric framework that distinguishes between generalized knowledge (residing in flat minima) and brittle memorization (residing in sharp minima) by probing the local curvature of the loss landscape via continuous embedding perturbations. Our theoretical analysis and extensive experiments across multiple open-weight models demonstrate that EPGS serves as a computationally efficient proxy for Hessian-based sharpness, achieving state-of-the-art performance in detecting high-confidence errors where both predictive uncertainty and static representation analyses fail. The code is publicly available at https://github.com/potato0o/Stubborn_Hallucinations.

## Impact Statement

This work aims to advance the field of Machine Learning by addressing a critical safety concern in Large Language Models: "Stubborn Hallucinations," where models generate factually incorrect information with high confidence and

stability. By providing a geometric framework (EPGS) to distinguish between robust knowledge and brittle memorization, this research has significant implications for the deployment of LLMs in high-stakes domains such as healthcare, legal analysis, and finance. In these fields, the inability of traditional uncertainty metrics to flag confident errors can lead to dangerous misinformation and decision-making failures.

Our method improves the interpretability and reliability of open-weight models, offering a mechanism to filter out entrenched errors that mimic factual knowledge. However, we acknowledge that our approach relies on white-box access to model gradients, which currently limits its application to open-source ecosystems and excludes closed-source API-based models. Additionally, while more efficient than full Hessian computation, the method introduces a computational overhead compared to standard inference, which entails increased energy consumption that practitioners must weigh against the necessity for high-fidelity verification. Ultimately, this work contributes to the broader goal of aligning model confidence with factual correctness, a prerequisite for trustworthy AI systems.

## Acknowledgement

We thank the anonymous reviewers for their constructive feedback and helpful comments. This work was supported by the Xi'an Jiaotong-Liverpool University Postgraduate Research Support (PGRS) grant FOSA2312001.

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

# A. Theoretical Derivations

In this section, we provide the rigorous mathematical derivation connecting Embedding-Perturbed Gradient Sensitivity (EPGS) to the Hessian spectrum of the loss landscape.

## A.1. Problem Formulation for Deep Architectures

Let the Large Language Model (LLM) be defined as a function $f_\theta : \mathbb{R}^{L \times d} \to \mathbb{R}^{L \times V}$, mapping a sequence of input embeddings $\mathbf{E}$ to logits. We decompose the network into two parts:

1. **The Network Body ($\Phi$):** Represents the composition of the first $N-1$ transformer layers. It maps input embeddings $\mathbf{E}$ to the final hidden state $\mathbf{H}_{N-1}$.

$$\Phi : \mathbb{R}^{L \times d} \to \mathbb{R}^{L \times d} \tag{8}$$

2. **The Final Block ($\mathcal{T}_{\text{last}}$):** Represents the $N$-th transformer layer (and subsequent norms/heads) parameterized by $\theta_{\text{last}}$. It maps the hidden state to the output.

$$\mathcal{T}_{\text{last}} : \mathbb{R}^{L \times d} \times \Theta_{\text{last}} \to \mathbb{R}^{L \times V} \tag{9}$$

We analyze the sensitivity of the loss $\mathcal{L}$ with respect to the parameters $\theta_{\text{last}}$ under a stochastic perturbation of the input embeddings.

## A.2. Propagation of Perturbation (The Jacobian Link)

Consider an isotropic Gaussian perturbation $\delta \sim \mathcal{N}(0, \sigma^2 \mathbf{I})$ added to the input embedding $\mathbf{E}$. Let $\mathbf{h} = \Phi(\mathbf{E})$ denote the hidden state input to the final block. Applying a first-order Taylor expansion to $\Phi$, the perturbation in the final hidden state, denoted as $\Delta \mathbf{h}$, is given by:

$$\Delta \mathbf{h} = \Phi(\mathbf{E} + \delta) - \Phi(\mathbf{E}) \approx \mathbf{J}_\Phi(\mathbf{E})\delta \tag{10}$$

where $\mathbf{J}_\Phi(\mathbf{E})$ is the Jacobian of the network body with respect to the input.

**Assumption A.1** (Lipschitz Continuity)**.** We assume the network body $\Phi$ is locally Lipschitz continuous with constant $K_\Phi$. This is a standard assumption in deep learning stability analysis, ensuring that bounded input noise does not result in exploding activation divergence for stable models:

$$\|\Delta \mathbf{h}\|_2 \leq K_\Phi \|\delta\|_2 \tag{11}$$

## A.3. Proof of Lemma 2

*Lemma 2 states that an input perturbation is locally equivalent to a parameter perturbation.*

*Proof.* We focus on the linear projection components of the final transformer block (e.g., the Feed-Forward Network or Attention output projection), denoted by weights $\mathbf{W} \in \theta_{\text{last}}$. The operation on the hidden state $\mathbf{h}$ is linear: $z = \mathbf{W}\mathbf{h}$.

Under the propagated noise $\Delta \mathbf{h}$, the perturbed activation is:

$$z_{\text{perturbed}} = \mathbf{W}(\mathbf{h} + \Delta \mathbf{h}) = \mathbf{W}\mathbf{h} + \mathbf{W}(\Delta \mathbf{h}) \tag{12}$$

We seek an induced parameter perturbation $\nu = \Delta \mathbf{W}$ such that the output on the *original* hidden state matches the perturbed output:

$$z' = (\mathbf{W} + \Delta \mathbf{W})\mathbf{h} = \mathbf{W}\mathbf{h} + (\Delta \mathbf{W})\mathbf{h} \tag{13}$$

Equating the deviation terms, we require $(\Delta \mathbf{W})\mathbf{h} = \mathbf{W}(\Delta \mathbf{h})$. Since this system is under-determined, we construct a rank-1 update utilizing the outer product:

$$\Delta \mathbf{W} = \frac{\mathbf{W}(\Delta \mathbf{h})\mathbf{h}^\top}{\|\mathbf{h}\|_2^2} \tag{14}$$

*Verification:*

$$(\Delta \mathbf{W})\mathbf{h} = \frac{\mathbf{W}(\Delta \mathbf{h})(\mathbf{h}^\top \mathbf{h})}{\|\mathbf{h}\|_2^2} = \mathbf{W}(\Delta \mathbf{h}) \tag{15}$$

Thus, there exists a parameter perturbation $\nu = \text{vec}(\Delta \mathbf{W})$ that functionally replicates the effect of the input noise $\delta$ at the layer output.

$\square$

### A.4. Proof of Theorem 3.4

*Theorem 3.4 asserts that the gradient norm under input noise is bounded by the Hessian spectral radius.*

*Proof.* Let $\theta^*$ be a local minimum such that $\nabla_{\theta_{\text{last}}} \mathcal{L}(\theta^*) \approx \mathbf{0}$. We examine the gradient of the loss with respect to the parameters of the last block, evaluated at the perturbed input $\mathbf{E} + \delta$. By Lemma 3.3, evaluating $\mathcal{L}$ at input $\mathbf{E} + \delta$ is locally equivalent to evaluating at parameters $\theta^* + \nu$.

We perform a second-order Taylor expansion of the loss around $\theta^*$:

$$\mathcal{L}(\theta^* + \nu) \approx \mathcal{L}(\theta^*) + \nu^\top \nabla \mathcal{L}(\theta^*) + \frac{1}{2} \nu^\top \mathbf{H} \nu \tag{16}$$

Taking the gradient with respect to $\nu$ (which corresponds to the gradient w.r.t parameters observed during the backward pass):

$$\nabla_{\theta_{\text{last}}} \mathcal{L}(\mathbf{E} + \delta) \approx \nabla \mathcal{L}(\theta^*) + \mathbf{H}\nu \approx \mathbf{H}\nu \tag{17}$$

The magnitude of this gradient vector, which constitutes the magnitude term of our EPGS score ($g_{\text{perturbed}}$), is bounded by the operator norm of the Hessian:

$$\|g_{\text{perturbed}}\|_2 \approx \|\mathbf{H}\nu\|_2 \leq \|\mathbf{H}\|_2 \|\nu\|_2 = \lambda_{\max}(\mathbf{H}) \|\nu\|_2 \tag{18}$$

Substituting $\nu = \Delta \mathbf{W}$ from Eq. 15 and bounding terms:

$$\|\nu\|_F = \|\Delta \mathbf{W}\|_F \leq \frac{\|\mathbf{W}\|_2 \|\Delta \mathbf{h}\|_2 \|\mathbf{h}\|_2}{\|\mathbf{h}\|_2^2} = \frac{\|\mathbf{W}\|_2 \|\Delta \mathbf{h}\|_2}{\|\mathbf{h}\|_2} \tag{19}$$

Finally, incorporating the Lipschitz bound from Assumption A.1 ($\|\Delta \mathbf{h}\|_2 \leq K_\Phi \|\delta\|_2$):

$$\|g_{\text{perturbed}}\|_2 \leq \underbrace{\lambda_{\max}(\mathbf{H})}_{\text{Sharpness}} \cdot \underbrace{\left( \frac{\|\mathbf{W}\|_2 K_\Phi}{\|\mathbf{h}\|_2} \right)}_{\text{Model Context}} \cdot \underbrace{\|\delta\|_2}_{\text{Noise}} \tag{20}$$

**Interpretation:** For a fixed noise magnitude $\|\delta\|_2$ and a converged model state (where latent representations $\mathbf{h}$ and weights $\mathbf{W}$ are stable), the gradient magnitude is dominated by $\lambda_{\max}(\mathbf{H})$.

1. **Stubborn Hallucinations:** Reside in sharp minima $\rightarrow$ Large $\lambda_{\max}$ $\rightarrow$ Large Gradient Spike.

2. **Robust Facts:** Reside in flat minima $\rightarrow$ Small $\lambda_{\max}$ $\rightarrow$ Negligible Gradient Spike.

This confirms that input sensitivity is a theoretically grounded proxy for the Hessian spectrum.

$\square$

## B. Algorithm of EPGS

We summarize the proposed EPGS framework in Algorithm 1. The process begins by isolating the target entity within the generated pseudo-label to focus the gradient signal on factual content. We then inject isotropic Gaussian noise into the input embeddings to actively probe the local curvature of the loss landscape. Finally, the detection score is computed by measuring the magnitude increase and directional divergence of the gradients in the last transformer block, effectively separating robust knowledge from brittle memorization.

---

**Algorithm 1** Embedding-Perturbed Gradient Sensitivity (EPGS)

---

**Require:** Pre-trained LLM $f_\theta$, Input Query $x$, Perturbation Scale $\sigma$, Stabilization Constant $\epsilon$
**Ensure:** Hallucination Score $S$
1: **Phase 1: Target Acquisition**
2: $\hat{y} \leftarrow \text{GreedyDecode}(f_\theta, x)$ {Get anchor label}
3: $y_{entity} \leftarrow \text{NER}(\hat{y})$ {Extract key entity}
4: $M \leftarrow \text{CreateMask}(\hat{y}, y_{entity})$ {Binary mask for entity tokens}
5: **Phase 2: Stochastic Embedding Perturbation**
6: $E \leftarrow \text{GetInputEmbeddings}(f_\theta, x)$
7: $\delta \sim \mathcal{N}(0, \sigma^2 I)$ {Sample Gaussian noise }
8: $E_{perturbed} \leftarrow E + \delta$
9: **Phase 3: Gradient Sensitivity Measurement**
10: $\mathcal{L}_{clean} \leftarrow \mathcal{L}_{masked}(f_\theta(E), \hat{y}, M)$
11: $g_{clean} \leftarrow \nabla_{\theta_{last}} \mathcal{L}_{clean}$
12: $\mathcal{L}_{perturbed} \leftarrow \mathcal{L}_{masked}(f_\theta(E_{perturbed}), \hat{y}, M)$
13: $g_{perturbed} \leftarrow \nabla_{\theta_{last}} \mathcal{L}_{perturbed}$
14: **Calculate EPGS Score**
15: $\text{CosSim} \leftarrow \frac{g_{clean} \cdot g_{perturbed}}{\max(\|g_{clean}\|_2, \epsilon) \max(\|g_{perturbed}\|_2, \epsilon)}$
16: $S \leftarrow \|g_{clean}\|_2 \cdot (1 - \text{CosSim})$ {Combines magnitude & direction }
17: **return** $S$

---

## C. Additional Experiment

### C.1. Empirical Validation of Hessian Sharpness

A core theoretical premise of our work (Hypothesis 3.2) is that stubborn hallucinations reside in sharp minima, characterized by a large Hessian spectral norm ($\lambda_{\max}$), whereas robust facts reside in flat minima. To provide a direct empirical link between theory and practice, we explicitly computed the top Hessian eigenvalue via power iteration.

We conducted this analysis on 50 robust facts and 50 stubborn hallucinations sampled from the TriviaQA dataset. The sample collection follows the methodology described in Section 5.3, ensuring that all selected instances maintain an output prediction probability exceeding $80\%$. Using the Llama-3-8B model, we observed a strong positive correlation between the proposed EPGS score and true Hessian sharpness. As illustrated in Figure 2, this relationship yields a Pearson correlation coefficient of $r = 0.855$ ($p < 10^{-28}$) and a Spearman rank correlation of $\rho = 0.862$. These results empirically validate that our computationally efficient, gradient-based metric serves as a high-fidelity proxy for the actual curvature of the loss landscape.

Crucially, this analysis substantiates our geometric definition of Stubborn Hallucinations. We demonstrate that entrenched factual errors consistently occupy sharper basins; specifically, the average $\lambda_{\max}$ for stubborn hallucinations is $2.5\times$ larger than that of generalized, robust facts. This empirical evidence firmly establishes that "confidently wrong" errors are topologically distinct from true knowledge, and that this geometric sharpness provides a reliable signal for detection.

### C.2. General Hallucinations Detection with SQuAD-F1

To verify the robustness of our evaluation framework, we replicate the detection benchmark from Table 1 utilizing SQuAD-F1 as the ground truth labelling criterion, as detailed in Table 5. Distinct from BERTScore, which quantifies soft semantic alignment, SQuAD-F1 enforces a rigorous penalty for deviations in lexical overlap between the generated response and the reference.

**Performance Analysis** While EPGS remains highly competitive, we observe a contraction in the performance margin compared to the BERTScore evaluation presented in Table 1. Notably, on the SQuAD dataset utilizing Llama-2-7B, EPGS (0.7109) is marginally outperformed by Effective Rank (0.7578) and Semantic Entropy (0.7511). We attribute this variation primarily to the lexical rigidity of the SQuAD-F1 metric. Since Large Language Models are stochastic and capable of diverse expression, a generated response may possess full semantic equivalence to the ground truth, for instance, through valid paraphrasing, while lacking exact lexical matching. Consequently, this phenomenon does not indicate a degradation in

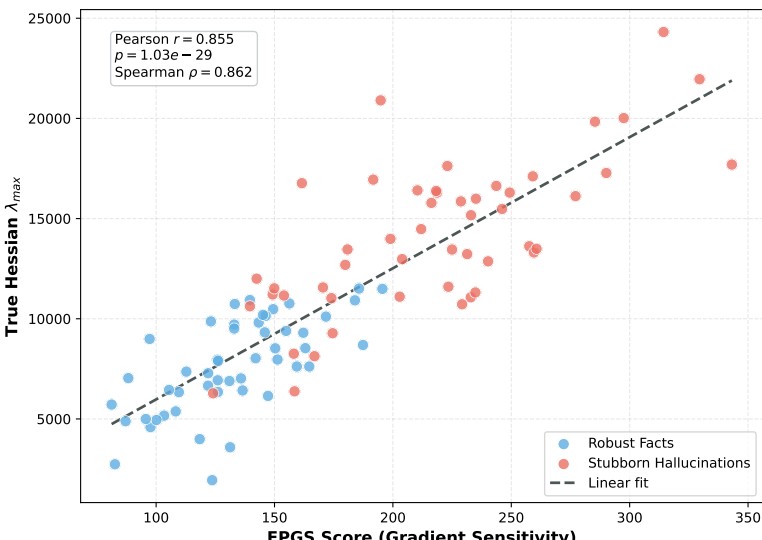

*Figure 2.* Correlation between the proposed EPGS score and the true top Hessian eigenvalue ($\lambda_{\max}$) computed via 15 power iteration on Llama-3-8B ("Stubborn" TriviaQA).

the method's intrinsic detection capability, but rather reflects a misalignment between rigid n-gram based metrics and the open-ended generative nature of modern LLMs.

## C.3. Extended Evaluations on Additional Architectures and Benchmarks

To further validate the universality of our geometric framework, we extend our evaluation to include additional modern LLM architectures and specialized hallucination benchmarks. Table 6 presents the detection performance on the Qwen-2.5-1.5B and 14B (Qwen et al., 2025) and Gemma-3-4B (Team et al., 2025) model families across the TriviaQA, SQuAD, and Natural Questions datasets. The results confirm that the topological distinction between flat facts and sharp hallucinations is not unique to the Llama or Mistral families, but is a fundamental property of transformer-based language models. EPGS consistently maintains state-of-the-art performance across these architectures, demonstrating particular robustness on the larger Qwen-2.5-14B model where it significantly outperforms both entropy-based and static representation baselines.

Furthermore, we evaluate our method on two challenging hallucination-specific datasets: TruthfulQA (Lin et al., 2022a), which probes the model's susceptibility to human falsehoods and misconceptions, and HaluEval (Li et al., 2023), a comprehensive hallucination evaluation benchmark. As detailed in Table 7, EPGS exhibits superior generalizability. Notably, on TruthfulQA, where models frequently output highly confident but factually incorrect mimicked falsehoods, static white-box methods like EigenScore and Effective Rank, suffer a severe performance collapse (AUROC $\approx 0.34 - 0.39$). In contrast, EPGS maintains a robust detection signal (AUROC $\approx 0.65 - 0.69$), successfully exposing the brittle, memorized nature of these falsehoods through gradient sensitivity. These extended evaluations underscore the reliability of geometric curvature as a universal discriminator for stubborn errors across diverse generative contexts.

## C.4. Ablation Study: Type of Perturbation

Our approach is grounded in the **Input-Parameter Isomorphism** (Lemma 3.3), which posits that a perturbation in the input embedding space is locally equivalent to a perturbation in the model parameters. Historically, the direct application of Gaussian noise to LLM embeddings has been viewed with skepticism, with prior research suggesting that continuous perturbations might destroy semantic structure compared to discrete tokens (Zhu et al., 2020). Consequently, the field has gravitated toward Discrete Semantic Prompting (e.g., paraphrasing) or *Structural Perturbations* (e.g., Monte Carlo Dropout) to induce uncertainty. In this work, we challenge this convention, hypothesizing that for gradient sensitivity analysis, continuous embedding perturbation is not only sufficient but serves as a high-fidelity proxy for parameter-space curvature.

The empirical results in Table 8 provide strong validation for this hypothesis and, by extension, confirm the correctness

*Table 5.* Comparison of General Hallucination Detection methods across Llama-2-7B, Llama-3-8B, and Mistral-7B-v0.1 using `SQuAD-F1` for labeling. We compare Black Box methods (LNE, PF, SE, DSE) and White Box methods (ER, ES) against our proposed method (EPGS). All values are AUROC scores (higher is better). Best results are in **bold**, second-best are underlined.

| Model | Dataset | Black Box | | | | White Box | | |
|---|---|---|---|---|---|---|---|---|
| | | LNE | PF | SE | DSE | ER | ES | EPGS (Ours) |
| Llama-2-7B | TriviaQA | 0.7004 | 0.6722 | 0.7790 | 0.7803 | 0.7909 | 0.7816 | **0.7997** |
| | SQuAD | 0.6912 | 0.5564 | 0.7511 | 0.7537 | **0.7578** | 0.7543 | 0.7109 |
| | NQ | 0.7182 | 0.6430 | 0.6860 | 0.7540 | 0.7421 | 0.7455 | **0.7571** |
| | SVAMP | 0.6641 | 0.7237 | 0.8517 | **0.8570** | 0.8530 | 0.8517 | 0.8186 |
| Llama-3-8B | TriviaQA | 0.6455 | 0.7306 | 0.7215 | 0.7205 | 0.7357 | 0.7328 | **0.7762** |
| | SQuAD | 0.6795 | 0.5870 | **0.7488** | 0.7428 | 0.7409 | 0.7342 | 0.7166 |
| | NQ | 0.6959 | 0.6635 | 0.7318 | 0.7297 | 0.7368 | 0.7353 | **0.7394** |
| | SVAMP | 0.6802 | 0.7563 | **0.8789** | 0.8753 | 0.8540 | 0.8677 | 0.8640 |
| Mistral-7B | TriviaQA | 0.6465 | 0.7306 | 0.7725 | 0.7689 | 0.7865 | 0.7756 | **0.8176** |
| | SQuAD | 0.6740 | 0.6416 | 0.7146 | **0.7187** | 0.7077 | 0.7100 | 0.7015 |
| | NQ | 0.7070 | 0.7421 | **0.7910** | 0.7908 | 0.7608 | 0.7643 | 0.7205 |
| | SVAMP | 0.6510 | 0.7228 | 0.8749 | 0.8801 | 0.8623 | 0.8602 | **0.8822** |

*Table 6.* Extended performance comparison of hallucination detection methods across Qwen-2.5-1.5B, Qwen-2.5-14B, and Gemma-3-4B. Best results are in **bold**, second-best are underlined.

| Model | Dataset | Black Box | | | | White Box | | |
|---|---|---|---|---|---|---|---|---|
| | | LNE | PF | SE | DSE | ES | ER | OUR |
| Qwen-2.5-1.5B | TriviaQA | 0.747 | 0.795 | 0.601 | 0.597 | **0.827** | 0.827 | 0.817 |
| | SQuAD | 0.562 | 0.527 | 0.562 | 0.561 | 0.612 | 0.614 | **0.711** |
| | NQ | **0.730** | 0.648 | 0.584 | 0.571 | 0.696 | 0.695 | 0.729 |
| Qwen-2.5-14B | TriviaQA | 0.591 | 0.760 | 0.590 | 0.565 | 0.794 | 0.795 | **0.817** |
| | SQuAD | 0.535 | 0.525 | 0.528 | 0.540 | 0.707 | 0.704 | **0.763** |
| | NQ | 0.480 | 0.683 | 0.651 | 0.645 | 0.779 | 0.771 | **0.817** |
| Gemma-3-4B | TriviaQA | - | - | - | 0.577 | 0.661 | 0.642 | **0.758** |
| | SQuAD | - | - | - | 0.629 | 0.643 | 0.645 | **0.676** |
| | NQ | - | - | - | 0.632 | 0.649 | **0.654** | 0.642 |

of Lemma 3.3. We compare EPGS against a comprehensive suite of perturbation strategies, including discrete prompts (Neutral, Active, Reasoning), Paraphrasing, and Inference-Time Dropout ($p = 0.3$). Crucially, Dropout represents a direct perturbation of the model's internal operators (parameter space). If Lemma 3.3 holds, our input-based EPGS should approximate the signal derived from Dropout. The results confirm this with remarkable consistency: EPGS matches or exceeds the performance of Dropout across 5 out of 6 experimental configurations. For instance, on Llama-3-8B (TriviaQA), EPGS achieves an AUROC of **0.8127**, outperforming Dropout (0.8028). Furthermore, EPGS significantly outperforms standard Paraphrasing (e.g., 0.7629 vs. 0.6985 on Llama-2-7B TriviaQA), suggesting that Gaussian noise provides a purer signal of local curvature than the uncontrolled semantic variance introduced by paraphrasing.

This performance equivalence serves as proof of the Lemma 3.3 (Input-Parameter Isomorphism): probing the loss landscape via the input embedding (EPGS) effectively recovers the same sensitivity signal as probing the weights directly (Dropout). Beyond theoretical validation, this finding has significant practical implications. While Dropout requires architectural intrusion such as forcing modifications to the forward pass that break standard inference optimizations (e.g., vLLM, fused kernels), on the other side, EPGS is non-intrusive. It operates solely on the input embeddings and final gradients, treating the model backbone as a fixed operator. By demonstrating that we can detect stubborn hallucinations with the same fidelity as Dropout but without touching the model internals, EPGS offers a superior trade-off between geometric rigor and deployment feasibility.

## C.5. Ablation Study: Importance of Named Entity Recognition (NER)

To validate the necessity and flexibility of Phase 1 (Target Acquisition), we evaluate the performance impact of our entity masking strategy. We compare our default `BERT`-based NER pipeline ("With NER") against two alternative configurations:

*Table 7.* Comparison of methods across Llama-2-7B, Llama-3-8B, and Mistral-7B on extended datasets (TruthfulQA and HaluEval). Best results are in **bold**, second-best are underlined.

| Model | Dataset | Black Box | | | | White Box | | |
|---|---|---|---|---|---|---|---|---|
| | | LNE | PF | SE | DSE | ES | ER | **OUR** |
| Llama-2-7B | TruthfulQA | 0.601 | 0.566 | 0.416 | 0.427 | 0.386 | 0.394 | **0.670** |
| | HaluEval | 0.628 | 0.530 | 0.526 | 0.535 | 0.656 | 0.656 | **0.738** |
| Llama-3-8B | TruthfulQA | 0.523 | 0.689 | 0.549 | 0.537 | 0.375 | 0.393 | **0.699** |
| | HaluEval | 0.585 | 0.615 | 0.528 | 0.518 | 0.653 | 0.651 | **0.770** |
| Mistral-7B | TruthfulQA | 0.580 | 0.647 | 0.419 | 0.443 | 0.353 | 0.342 | **0.658** |
| | HaluEval | 0.628 | 0.592 | 0.532 | 0.529 | 0.682 | 0.683 | **0.755** |

*Table 8.* Ablation study comparing different perturbation strategies across Llama-2-7B and Llama-3-8B. We compare Prompt Perturbation methods (Empty, Neutral, Active, Reasoning, Noise), Paraphrasing, and Dropout against our proposed Embedding Perturbation (EPGS). All values are AUROC scores. The best performance in each column is highlighted in **bold**.

| Type | Sub-type | Prompt Content | Llama-2-7B | | | Llama-3-8B | | |
|---|---|---|---|---|---|---|---|---|
| | | | Trivia | SQuAD | NQ | Trivia | SQuAD | NQ |
| Prompt Perturb | Empty | Space (Empty Prompt) | 0.7365 | 0.6977 | 0.7026 | **0.8147** | 0.7601 | 0.7585 |
| | Neutral | "Please answer this:" | 0.7629 | 0.6924 | 0.6951 | 0.8127 | 0.7742 | **0.7705** |
| | | "Simply state the answer:" | 0.7620 | 0.7183 | 0.6947 | 0.8126 | 0.7645 | 0.7702 |
| | Active | "You are a strict, factual encyclopedia." | **0.7742** | **0.7507** | 0.6947 | 0.8053 | **0.7847** | 0.7702 |
| | Reasoning | "Let's think step by step. Analyze the question first, then state the answer." | 0.7703 | **0.7507** | 0.6947 | 0.8122 | **0.7847** | 0.7702 |
| | Noise | "I just drank a coffee." | 0.7603 | 0.7085 | **0.7479** | 0.8093 | 0.7603 | 0.7550 |
| | | "What is the capital of France?" | 0.7550 | 0.7085 | **0.7479** | 0.7978 | 0.7603 | 0.7550 |
| Paraphrase | | *(N/A)* | 0.6985 | 0.6464 | 0.6963 | 0.7622 | 0.6828 | 0.7300 |
| Dropout ($p = 0.3$) | | *(N/A)* | 0.7608 | 0.6923 | 0.6863 | 0.8028 | 0.7832 | 0.7680 |
| **Embedding Perturb (EPGS)** | | *(N/A)* | 0.7629 | 0.6924 | 0.6951 | 0.8127 | 0.7742 | **0.7705** |

an "LLM (Llama-3-8B)" approach where the LLM is prompted to identify the key entities, and a "Without NER" baseline where the gradient sensitivity score $\mathcal{S}$ is computed over the entire generated sequence $\hat{y}$, including syntactic fillers and stop words.

The results, presented in Table 9, demonstrate that isolating the factual entity is crucial for detection accuracy. We observe a consistent and significant performance degradation when the mask is removed entirely, with the AUROC dropping by as much as 8.2% (Llama-2-7B on NQ). Our theoretical framework explains this through the mechanism of *Gradient Signal Dilution*. While stubborn hallucinations are characterized by sharp local curvature (producing large gradient spikes), syntactic tokens (e.g., "The", "is", "was") typically correspond to well-learned, highly probable structural patterns residing in flat, stable regions of the loss landscape. When the gradient norm is averaged over the entire sequence, these negligible gradients from stable tokens dilute the high-magnitude signal originating from the brittle, hallucinated entity, effectively lowering the signal-to-noise ratio of the detection metric.

This dilution effect explains the variance in performance drops across datasets. On TriviaQA, where responses are often concise, entity-centric phrases (low verbosity), the difference between masking and not masking is marginal. Conversely, on SQuAD and Natural Questions (NQ), where the model generates complete sentences containing a higher ratio of filler words to entities, the removal of the mask causes a significant collapse in performance.

Crucially, the strong performance of the "LLM (Llama3-8B)" configuration confirms that our method's success is not an artifact of the specific BERT NER model, but rather a fundamental property of the underlying geometric signal. LLM-based masking remains highly robust, even outperforming the standard NER pipeline on Llama-2-7B for SQuAD and NQ. This establishes that the geometric signal is intrinsic to the factual content itself; as long as a masking mechanism successfully

*Table 9.* Ablation study on the importance of Named Entity Recognition (NER). The best results are highlighted in **bold**.

| Configuration | Llama-2-7B | | | Llama-3-8B | | |
|---|---|---|---|---|---|---|
| | Trivia | SQuAD | NQ | Trivia | SQuAD | NQ |
| With NER | **0.7629** | 0.6924 | 0.6951 | **0.8127** | **0.7742** | **0.7705** |
| LLM (Llama3-8B) | 0.7322 | **0.7691** | **0.7516** | 0.7890 | 0.7375 | 0.7385 |
| Without NER | 0.7570 | 0.6334 | 0.6127 | 0.8068 | 0.7580 | 0.7387 |

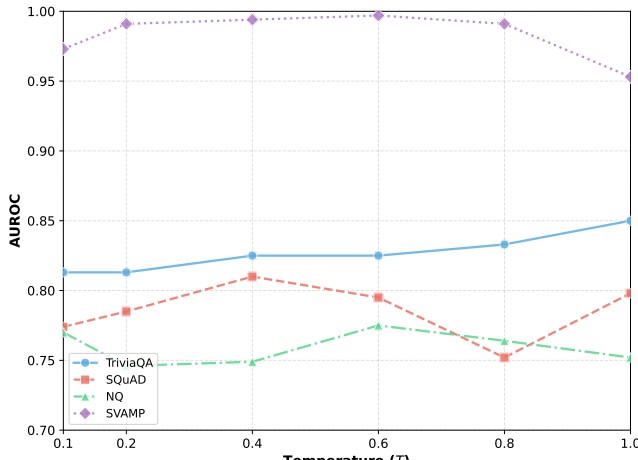

*Figure 3.* Effect of LLM decoding temperature ($T$) on EPGS hallucination detection performance (Llama-3-8B). The metric demonstrates high stability across the stochastic spectrum, maintaining robust detection capabilities regardless of whether generation is near-deterministic ($T = 0.1$) or highly stochastic ($T = 1.0$).

isolates the semantic focal point of the generation to prevent syntactic dilution, EPGS can accurately detect the geometric instability characteristic of stubborn hallucinations.

### C.6. Ablation Study: Sensitivity to Decoding Temperature

In our primary evaluation setup, we utilized a low-temperature configuration ($T = 0.1$) to approximate greedy decoding. This decision was designed to anchor our analysis on the model's most likely reasoning path, which strongly aligns with the definition of Stubborn Hallucinations (errors characterized by consistency and high confidence). However, verifying the robustness of a detection metric under stochastic decoding conditions is crucial for real-world deployment.

To evaluate this, we conducted an ablation study using the Llama-3-8B model across a broad spectrum of temperatures $T \in [0.1, 1.0]$. The results, presented in Figure 3, demonstrate that EPGS remains remarkably robust across all evaluated datasets. The detection performance (AUROC) fluctuates only marginally as the temperature increases, with higher temperatures occasionally yielding slight performance improvements (e.g., reaching 0.850 on TriviaQA at $T = 1.0$).

This stability confirms that EPGS successfully captures the underlying geometric sharpness of the local loss basin, regardless of whether that specific basin was reached via deterministic greedy decoding or stochastic sampling. Notably, on reasoning-heavy tasks like SVAMP, EPGS maintains near-perfect separation (AUROC $> 0.95$) across the entire temperature spectrum. These results empirically validate that the geometric curvature probed by EPGS provides a robust, decoding-independent signal for hallucination detection.

### C.7. Ablation Study: Calibration and Noise Ratio

To provide a normalized interpretation of our embedding perturbation, we analyze the scale of the injected noise relative to the original model representations. Table 10 details the ratio of the aggregate noise norm ($\|\delta\|_2$) to the original embedding norm ($\|\mathbf{E}\|_2$) on the TriviaQA dataset.

Due to the extremely high dimensionality of the embedding space ($d \geq 4096$), our default perturbation magnitude ($\epsilon = 0.1$)

results in an aggregate stochastic shift that is significantly larger than the original embedding magnitude across all models. For instance, on Llama-3-8B, the aggregate noise norm is over $14\times$ larger ($1414\%$) than the clean embedding norm.

Remarkably, despite these massive shifts in the absolute latent space, the detection performance (AUROC) of EPGS remains exceptionally stable. As previously demonstrated in Table 4, performance fluctuates by less than 2% across noise magnitudes $\epsilon \in [0.001, 1]$. This reveals a profound insight into the topology of the loss landscape: the geometric "sharpness" we probe is not a microscopic, local artifact, but a fundamental, macroscopic topological feature.

The narrow, steep-walled basins characteristic of stubborn hallucinations are so dominant that they define the local landscape even under high-magnitude stochastic stress. Conversely, the flat basins of robust facts are wide enough to absorb these massive perturbations without inducing significant gradient spikes. This extreme robustness confirms that EPGS requires virtually no per-model hyperparameter tuning; the geometric distinction between facts and stubborn errors persists across several orders of magnitude of applied noise.

*Table 10.* Ratio of the aggregate noise norm ($\|\delta\|_2$) to the original embedding norm ($\|\mathbf{E}\|_2$) on the TriviaQA dataset across different noise magnitudes ($\epsilon$). Due to high dimensionality, the aggregate perturbation can exceed the original embedding magnitude by orders of magnitude.

| $\epsilon$ | Llama-2-7B | Llama-3-8B | Mistral-7B |
|---|---|---|---|
| 0.001 | 8.46% | 14.13% | 47.11% |
| 0.01 | 84.55% | 141.28% | 471.06% |
| 0.1 | 846.00% | 1414.00% | 4714.85% |

## C.8. Ablation Study: Computational Efficiency and Complexity Analysis

To assess practical viability, we analyze EPGS's computational overhead. Unlike sampling-based methods that require multiple generated trajectories (Farquhar et al., 2024), EPGS operates on a single generation. Its overhead stems from dual forward and backward passes (clean and perturbed). Crucially, by restricting gradient computation to the final transformer block ($\theta_{\text{last}}$), the backward-pass cost is independent of model depth. This $\mathcal{O}(1)$ depth scaling avoids the prohibitive $\mathcal{O}(P^2)$ complexity of full Hessian or spectral approximations.

Empirical results in Table 11 substantiate this efficiency. On TriviaQA, while the backward passes add moderate latency compared to forward-only metrics like LNE, EPGS is an order of magnitude faster than white-box spectral methods (ER and ES). For instance, ER and ES require upwards of 2800s on Llama-2-7B, whereas EPGS requires only 263s.

Furthermore, EPGS exhibits highly favorable scaling on complex, long-form outputs. Transitioning from short answers (TriviaQA) to long generations (TruthfulQA) on Llama-3-8B, the latency of forward-only LNE increases by 91% (113s $\rightarrow$ 215s) due to compounded autoregressive costs. In contrast, EPGS latency grows by only 27% (274s $\rightarrow$ 349s). This shrinking relative overhead is driven by Selective Factual Probing (Phase 1), which ensures computational costs scale strictly with key entity claims rather than the total word count.

Finally, storing final-block gradients requires only a marginal, sub-linear increase in VRAM (13.95 GB $\rightarrow$ 15.59 GB for Llama-2-7B). Consequently, EPGS remains fully deployable on single consumer-grade GPUs (RTX 4090 as used in this paper), securing deep geometric insights without prohibitive resource demands.

# D. Qualitative Result

## D.1. Qualitative Comparison: EPGS vs. Entropy Metrics

To provide deeper intuition into why predictive stability fails while geometric curvature succeeds on stubborn hallucinations, we present a side-by-side qualitative comparison in Figure 4. We contrast our EPGS against leading entropy-based metrics: Length-Normalized Entropy (LNE) and Discrete Semantic Entropy (DSE).

A fundamental limitation of output-dependent methods like LNE and DSE is their inherent hypersensitivity to "paraphrase noise." Minor syntactic variations in the generation trajectory, such as changes in capitalization, the inclusion of filler words, or the use of synonyms (e.g., *"Golf"* vs. *"golf"* vs. *"A good golf player"*), can artificially inflate entropy scores despite the semantic meaning remaining entirely stable. This syntactic volatility causes traditional metrics to fluctuate wildly, rendering them unreliable for isolating deeply entrenched factual errors.

*Table 11.* Efficiency comparison of different hallucination detection methods. We report Latency and Peak VRAM across multiple models on TriviaQA, alongside Latency on TruthfulQA.

| Type | Method | TriviaQA | | | | | | TruthfulQA |
|---|---|---|---|---|---|---|---|---|
| | | Latency (seconds) | | | Peak VRAM (GB) | | | Latency (seconds) |
| | | Llama-2-7B | Llama-3-8B | Mistral-7B | Llama-2-7B | Llama-3-8B | Mistral-7B | Llama-3-8B |
| EPGS (Ours) | | 263.26 | 273.91 | 294.44 | 15.59 | 18.04 | 16.74 | 348.75 |
| Forward only | LNE SE DSE | 123.29 | 112.96 | 116.97 | 13.95 | 14.55 | 13.97 | 215.35 |
| White box | ER ES | 2846.51 | 2776.18 | 2783.12 | 14.90 | 17.64 | 15.86 | 3605.59 |

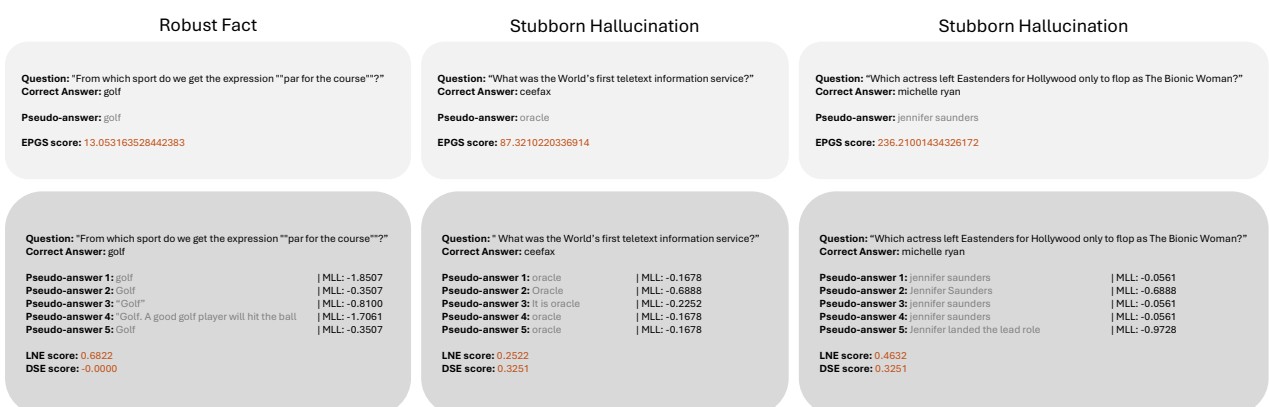

*Figure 4.* Side-by-side qualitative comparison of EPGS against entropy-based metrics (LNE and DSE).

By contrast, EPGS bypasses this "fluency bottleneck" entirely by operating on the dense latent representations and computing the gradient sensitivity in the final transformer block. As demonstrated in Figure 4, when the model encounters a stubborn hallucination (a sharp minimum), EPGS correctly identifies a massive 6–18× increase in gradient magnitude. This confirms that EPGS effectively captures the underlying geometric "sharpness" of memorized falsehoods, providing a robust detection signal even when the surface-level output probabilities remain overconfident and stable.

### D.2. Qualitative Sample: TriviaQA "Stubborn" subset on Meta-Llama3-8B

Figure 5 provides further qualitative examples drawn specifically from the Stubbornness Subset of TriviaQA using Meta-Llama3-8B.

## E. Further Discussion

### E.1. Validity of Hallucination Labelling

We acknowledge that our labelling strategy relies on determining factual correctness via reference comparison rather than human evaluation. While some definitions separate "reasoning errors" from "hallucinations", we posit that in the context of knowledge retrieval, this distinction is topologically irrelevant. Whether caused by a failure in logic or a failure in retrieval, a Stubborn Hallucination is defined by the model's convergence to an incorrect state with high stability.

Standard static benchmarks fail to capture this phenomenon because they decouple the error from the generation process. Because hallucination is not a property of the dataset but a property of the model-data interaction, there is no single "Hallucination Set" that contains the correct pattern of error for every LLM. For example, Llama-2 may hallucinate on a specific entity in TriviaQA while Mistral-7B retrieves it correctly; a static label would fail to distinguish these states. Therefore, our method of hallucination labelling using `BertScore` or `SQuad-F1` treat the model's own incorrect, stable

> **Question:** The Flying Pickets were a British vocal group who had Christmas no1 hit in 1983. What was the title of the song.
> **Correct Answer:** only you
> **Pseudo-answer:** only you
> **Curvature Score:** 11.771678924560547

> **Question:** What was the World's first teletext information service?
> **Correct Answer:** ceefax
> **Pseudo-answer:** oracle
> **Curvature Score:** 87.3210220336914

> **Question:** What is the lowest level of the Earth's atmosphere?
> **Correct Answer:** troposphere
> **Pseudo-answer:** troposphere
> **Curvature Score:** 19.5494384765625

> **Question:** What certificate is often earned after graduating high school?
> **Correct Answer:** elementary school education certificate
> **Pseudo-answer:** GED
> **Curvature Score:** 102.38825225830078

> **Question:** What is the number of Constituency MSPs?
> **Correct Answer:** 73
> **Pseudo-answer:** 73
> **Curvature Score:** 3.849358558654785

*Figure 5.* Qualitative examples showing the Question, Correct Answer, Pseudo-answer, and the calculated Curvature Score.

generations as the positive class for hallucination is necessary. It allows us to isolate the specific "sharp minima" that correspond to that specific model's memorized errors, providing a more rigorous test bed for the EPGS metric than pre-fabricated adversarial samples.

### E.2. Theoretical Validity of the Curvature Hypothesis

A core premise of our work (Hypothesis 3.2) is that robust factual knowledge resides in "flat minima" of the loss landscape, while stubborn hallucinations occupy "sharp minima," a view grounded in classical generalization theory (Hochreiter & Schmidhuber, 1997; Keskar et al., 2017). However, we must address a prominent counter-narrative in deep learning theory: the work of (Dinh et al., 2017), which argues that sharp minima can generalize. They demonstrate that Hessian-based sharpness measures are not invariant to re-parameterization; for strictly non-negative homogeneous networks, one can arbitrarily rescale weights to increase the spectral norm of the Hessian without altering the function's output. Theoretically, this implies that a sharp minimum can be mathematically equivalent to a flat one, challenging the universality of the link between curvature and generalization. While this critique is mathematically sound regarding architectural invariance, we argue that it does not invalidate our hypothesis within the specific context of analyzing fixed, pre-trained Large Language Models.

The invariance argument put forth by (Dinh et al., 2017) relies on the ability to re-parameterize the network post-hoc to manipulate the geometry. In contrast, our method operates on fixed pre-trained checkpoints where the parameterization is frozen and the coordinate system is determined by the specific trajectory of the optimizer (e.g., Adam). Within this fixed coordinate system, extensive empirical studies have shown that sharpness measures remain highly predictive of generalization error (Jiang* et al., 2020). By holding the architecture and weights constant, we are not comparing sharpness across different models or parameter scales, but rather probing the relative curvature differences between specific data points (facts vs. hallucinations) within the same model. In this "inference-only" setting, the geometric distinction serves as a valid proxy for the robustness of the learned representation.

Furthermore, we posit that the distinction between facts and stubborn hallucinations is rooted in the mechanism of information storage described by information geometry. Generalized knowledge is characterized by feature redundancy, where the model relies on multiple distributed circuits to encode a concept, resulting in a wide basin of attraction where individual parameter perturbations are dampened. Conversely, stubborn hallucinations mimic the behavior of memorized noise. Theoretical work by (Arpit et al., 2017; Feldman, 2020) suggests that while deep networks learn simple, generalized patterns in stable regions, they "cram" noisy or outlier data into high-curvature regions to satisfy the training objective. In this context, "sharpness" is not merely a geometric artifact subject to re-parameterization, but a signature of overfitting to a heuristic without the support of generalized features.

Finally, while the debate regarding parameter-space sharpness is nuanced, our method explicitly measures input-embedding sensitivity, which offers a semantically unambiguous metric for LLMs. If a prediction is stable only within a microscopic radius of the input embedding (high sensitivity), the model fails to possess semantic robustness regardless of the underlying weight scaling. The EPGS score reflects this lack of margin. By linking this input instability to parameter curvature via the Jacobian (Lemma 3.3), we utilize sharpness not as an absolute measure of generalization capacity, but as a differential diagnostic tool to separate robust concepts from brittle memorization.

