# OpenReview forum: "From Flat Facts to Sharp Hallucinations: Detecting Stubborn Errors via Gradient Sensitivity"
_ICML.cc/2026/Conference — ICML 2026 regular_

### Official Review · Reviewer_fG6S · 2026-03-05

**Soundness:** 2
**Presentation:** 2
**Significance:** 2
**Originality:** 3
**Overall Recommendation:** 3
**Confidence:** 3

**Summary:**

This paper proposes Embedding-Perturbed Gradient Sensitivity (EPGS), a white-box detector for LLM hallucinations, particularly “stubborn” hallucinations where the model is confidently wrong. The core idea is to inject Gaussian noise into the input embeddings and measure the change in gradients of the final transformer block relative to a masked, entity-focused pseudo-label; the resulting score combines gradient magnitude and angular divergence. The authors provide a geometric motivation (sharp vs. flat minima), a first-order equivalence argument between embedding perturbations and an induced parameter perturbation in the last block, and experiments on QA and arithmetic reasoning benchmarks where EPGS reportedly outperforms black-box entropy-based and white-box representation-based baselines, including on a curated “stubborn subset.”

**Compliance With Llm Reviewing Policy:**

Affirmed.

**Final Justification:**

It addresses my concerns. Considering the contribution of this paper, I will keep my score.

**Key Questions For Authors:**

See weaknesses above.

**Limitations:**

See weaknesses above.

**Strengths And Weaknesses:**

**Strengths:**
1. The paper brings a geometric perspective, linking curvature of the loss landscape to hallucination detectability.
2. Theoretical claims (Definition 3.1, Lemma 3.3, Theorem 3.4) link the gradient response to Hessian sharpness.
3. Experiments demonstrate robust and significant gains over both black-box and white-box baselines.
4. Useful ablations are included: noise magnitude sensitivity and gradient extraction location.
5. The paper is well-structured, with a clear pipeline.

** Weaknesses:**
1. The curvature hypothesis is intuitively motivated by generalization theory, but the connection to factual correctness during inference is not entirely clear to me. It would be useful to better explain why stubborn hallucinations are expected to manifest as sharp minima instead of arising from other mechanisms.
2. The analysis links parameter-space perturbations to input perturbations through *Lemma 3.3*, which appears to rely on local approximations around $\theta^{*}$. It would be helpful to discuss the extent to which this assumption remains valid in practice, particularly in the presence of masking and nonlinearities.
3. The cosine-similarity term can be numerically unstable when gradients are small. No stabilization (e.g., ε in norms) is documented.
4. The paper does not compare with recent geometry-/spectral-based works for hallucination detection.

---

> ### Author Rebuttal · Authors · 2026-03-28
>
> We sincerely thank the reviewer for highlighting our theoretical contribution, which links the gradient response to Hessian sharpness.
>
> **A1: Why stubborn hallucinations manifest as sharp minima**
> We define "stubborn hallucination" as factually incorrect predictions that remain highly confident and stable in the output space. To explain why these manifest as sharp minima, we draw on information geometry and memorization theory [1,2].
>
> Deep networks naturally learn generalized patterns (true facts) by relying on multiple distributed, redundant circuits. This feature redundancy inherently creates wide, stable basins of attraction (flat minima). Conversely, stubborn hallucinations arise when a model overfits to sparse, noisy, or spurious training data (rote memorization). Lacking redundant feature support, the network "crams" this isolated, incorrect mapping into a narrow, high-curvature basin simply to minimize training loss.
>
> Because this memorized mapping is structurally brittle, it is highly sensitive to local shifts. During inference, even a microscopic
> continuous perturbation to the input forces the activation state up the steep walls of this narrow loss basin. This topological constraint causes the gradient to spike dramatically (as visualized in Figure 1), allowing EPGS to successfully detect the hallucination despite its stable output probability.
>
> To move beyond intuition, we computed the top Hessian eigenvalue ($\lambda_{max}$) via power iteration on a representative subset of TriviaQA using Llama-3-8B. Our results (**Figure A** on the anon site) reveal an exceptionally strong correlation between the EPGS score and true Hessian sharpness (Pearson $r = 0.855$, $p < 10^{-28}$). Crucially, we found that stubborn hallucinations reside in significantly sharper basins, with an average $\lambda_{max}$ that is $2.5\times$ larger than that of robust facts. This confirms that entrenched errors are indeed characterized by sharp minima in modern LLMs.
>
> **A2: Lemma 3.3 local approximation vs nonlinearities**
> We acknowledge that Lemma 3.3 relies on a first-order Taylor approximation. While Transformer architectures employ global nonlinearities (GeLU, Softmax), foundational theoretical work on wide neural networks [3] mathematically demonstrates that highly overparameterized models behave as locally linear systems within a bounded neighborhood ($\epsilon$). Because the activation patterns of these smooth functions remain largely stable within a microscopic perturbation radius, the first-order linear approximation is rigorously justified.
>
> Crucially, our results validate that this local linearity holds in practice for LLMs: EPGS demonstrate remarkable performance stability across several orders of magnitude in noise scale (Table 4). Furthermore, the strong performance parity between our input-embedding perturbation (EPGS) and direct parameter-space perturbation (Dropout, Table 6) empirically confirms that input noise serves as a high-fidelity proxy for parameter-space curvature, successfully propagating through deep nonlinearities and gradient masking.
>
> **A3: Numerical instability**
> We appreciate this precise observation. In our implementation, we utilize `torch.nn.functional.cosine_similarity`, which inherently applies a stabilization constant ($\epsilon = 1\mathrm{e}{-8}$) to the denominator to prevent division by zero. We will explicitly document the $\epsilon$ stabilization term in Eq. 7 in the revision.
>
> **A4: Comparisons with recent geometry/spectral works**
> We respectfully point out that our evaluation already includes EigenScore and Effective Rank. These currently represent the SOTA for spectral and geometric methods operating on hidden states (representation-space geometry).
> To best of our knowledge, our work is the first to bypass logits / representations entirely and frame hallucination detection through the geometry of the loss landscape (parameter-space curvature/geometry). If the reviewer had a specific recent work in mind that operates in this space, we would be more than happy to discuss it and include it in our literature review.
>
> Anon Site: https://anonymous.4open.science/r/ICML_rebuttal-61FB/
>
> [1] V Feldman "Does learning require memorization? a short tale about a long tail." STOC 2020
>
> [2] Arpit, D., et al. "Closer look at memorization in deep networks." ICML 2017
>
> [3] Lee, J., et al. "Wide Neural Networks of Any Depth Evolve as Linear Models Under Gradient Descent." NeurIPS 2019

---

> > ### Author Rebuttal · Reviewer_fG6S · 2026-04-03
> >
> > Thanks for the authors' response. I've read the rebuttal and other reviewer's comments. Considering the other reviewer's comments and the overall quality of this paper, I perfer to keep my score.

---

> > > ### Author Response · Authors · 2026-04-04
> > >
> > > We sincerely appreciate your continued engagement with our rebuttal and are very glad to hear that your specific concerns have been fully resolved. We also note your consideration of the other reviews and the paper's overall quality in your decision to maintain your score.
> > >
> > > To provide further context on the overall quality and broader impact of our work, especially in light of the cross-reviewer discussions you mentioned. We would like to briefly synthesize our core contributions:
> > >
> > > - **Novel Geometric Perspective:** To the best of our knowledge, we are the first to explain hallucinations through a geometric lens. As detailed in our discussions with `Reviewers A7BD, A1`, we demonstrated a strong correlation between the EPGS score and true Hessian sharpness. This firmly confirms our hypothesis that stubborn hallucinations are characterized by sharp minima.
> > >
> > > - **Robust Gradient Sensitivity:** We introduce gradient sensitivity (EPGS), which successfully bypasses the "fluency bottlenecks" inherent in output-variation methods (LNE, PF, DSE). This yields a highly robust signal that is effectively immune to syntactic noise.
> > >
> > > - **Favorable Computational Scaling:** Regarding computational efficiency, our method scales remarkably well. As generation length increases from TriviaQA to TruthfulQA, the latency of forward-only inference (LNE) increases by ~91% (112.96s to 215.35s) due to the multiple forward passes required. In contrast, EPGS latency increases by only ~27% (273.91s to 348.75s). While LNE maintains a lower absolute baseline, this demonstrates that the relative computational overhead of EPGS actually shrinks significantly as generation length increases.
> > >
> > > We believe these theoretical insights and robust empirical results represent a strong step forward for the community. We hope this concise summary further clarifies the fundamental value of our work, and we kindly ask if you might reconsider your overall assessment of the paper's quality in light of these clarified strengths.

---

### Official Review · Reviewer_A7BD · 2026-03-09

**Soundness:** 3
**Presentation:** 3
**Significance:** 3
**Originality:** 3
**Overall Recommendation:** 5
**Confidence:** 3

**Summary:**

The paper argues that robust factual knowledge in LLMs tends to lie in regions where the loss landscape is locally flat, while “sharp” regions are associated with “stubborn hallucinations”, i.e., generations that are consistent yet incorrect, plausibly due to sparse or insufficient training evidence. Building on this intuition, the authors propose a white box detection method for stubborn hallucinations that relies on gradient access to score whether a given generation is likely hallucinated. The approach is conceptually grounded in loss sharpness and operationalized via a gradient sensitivity style signal.

**Compliance With Llm Reviewing Policy:**

Affirmed.

**Final Justification:**

Thank you for the rebuttal. The direct validation against the top Hessian eigenvalue addresses my main concern and significantly strengthens the theory-to-practice link. The authors also addressed my other questions clearly and concretely. I still think the entity-centric scope should remain clearly stated as a limitation, and the final version should better clarify the interpretation of the large perturbation norms. Overall, I am positively updated by the rebuttal, maintain my support for acceptance, and will increase my score.

Minor presentation note: in the figure, please align g_clean horizontally as in Phase 1 for visual consistency.

**Key Questions For Authors:**

* Datasets: Did you consider running on TruthfulQA or HaluEval QA for additional coverage?


* Baselines: Could you add (even if only as reference) comparisons or discussion relative to other hallucination evaluator classes, such as internal state based detectors and verifier or search augmented evaluators? Even if not optimized for best numbers, it would contextualize strengths and limitations.


* NER dependence: Could NER be replaced by a prompt based mechanism (e.g., asking the model to output the key spans/entities to score), to reduce pipeline brittleness?


* Calibration: What is the per model calibration cost? Which hyperparameters typically need re tuning when changing the model?


* Delta scaling: Can you clarify how δ relates to the embedding norm (is it small/large relative to ||E||)? Ablations on absolute values are helpful, but a normalized interpretation would improve portability.


**To increase my reviewer score**, the authors should strengthen the theory to practice link by directly validating their sharpness proxy (e.g., correlating it on a small model with an explicit top Hessian eigenvalue approximation via power iteration) and broaden the evidence with side by side qualitative examples plus additional benchmarks like TruthfulQA or HaluEval.

**Limitations:**

Entity centric design: It is unclear how well the method works when answers contain no clear entities (e.g., short, on point responses constrained by prompt, numbers, or generic statements). More evidence or analysis for non entity outputs would improve confidence in generality.

**Strengths And Weaknesses:**

Strengths:
* Clear motivation and transparency: Strong framing of “stubborn hallucinations” and why standard uncertainty signals can miss them, plus an explicit Limitations section.

* Simple and practical (with white box access): No extra large model training or heavy pipeline, seems feasible if gradients are available.


* Solid empirical results: Reported gains look consistent across the evaluated settings and baselines.


* Easy to follow, original framing: Clear writing and a memorable “flat facts vs sharp hallucinations” story; it also suggests (in principle) a distinction between stubborn and transient hallucinations, though transient cases are not evaluated.


* High relevance and potential impact: Hallucination detection is central for GenAI reliability, and the sharpness perspective is a useful new angle.


Weaknesses:

* The theoretical justification seems to rely on strong approximations. A more direct validation would strengthen the theory to practice link, e.g., on a small model, compare the proposed sharpness proxy with an explicit approximation of the top Hessian eigenvalue (via power iteration), potentially even restricted to subspaces (projections/LoRA/heads).


* I would have liked to see qualitative, side by side examples comparing this detector to other methods on the same prompts, especially for “stubborn” cases. This would help interpret what the score is capturing and where it fails.


* The main figure feels generic and does not add much intuition. A more method specific figure (e.g., a phase style pipeline that visually walks through the scoring steps) would make the approach easier to understand.


* Minor typo: line 131, column 2 “A Facts”.

---

> ### Author Rebuttal · Authors · 2026-03-28
>
> We thank the reviewer for the positive feedback on our "stubborn hallucinations" framework and the novelty of the EPGS signal.
>
> **A1: Direct validation of sharpness**
> To strengthen the link between theory and practice, we computed the top Hessian eigenvalue ($\lambda_{max}$) via power iteration on stubborn subset of TriviaQA using Llama-3-8B. Our results (**Figure A** in anon site) reveal a strong correlation between EPGS score and true Hessian sharpness (Pearson $r = 0.855$, $p < 10^{-28}$). Notably, stubborn hallucinations reside in sharper basins, with an average $\lambda_{max}$ that is $2.5\times$ larger than that of robust facts. This confirms our hypothesis that entrenched errors are characterized by sharp minima.
>
> **A2: Methodology pipeline and qualitative examples**
> We thank the reviewer for the constructive critique of our visualization. While original Figure 1 intuitively demonstrate the geometry of stubborn hallucination, we have added a new Methodology Pipeline Figure (**Figure B** in anon site) that walks through the three phases of EPGS: (1) Target Acquisition, (2) Gradient Probing via Perturbation, and (3) Scoring.
>
> Additionally, we include a Side-by-Side Qualitative Comparison (**Figure C** in anon site) comparing EPGS with entropy-based metrics (LNE, DSE). We highlight that output-dependent methods like LNE and DSE are inherently sensitive to "paraphrase noise"—where minor variations in phrasing (e.g., capitalization or synonyms) can artificially inflate entropy despite semantic stability. By utilizing gradient sensitivity in the last transformer block, EPGS bypasses these surface-level probability fluctuations. As observed, our method correctly identifies a massive 6–18$\times$ increase in gradient magnitude for stubborn hallucinations, capturing the underlying geometric "sharpness" even when output probabilities remain overconfident.
>
> **A3: Additional benchmarks and evaluators**
> As requested, we evaluated EPGS on TruthfulQA and HaluEval. EPGS consistently outperforms existing white-box and black-box baselines:
>
> | **Dataset**    | **Model** | **LNE** | **PF** | **ER** | **EPGS** |
> | -------------- | --------- | ------- | ------ | ------ | -------- |
> | **TruthfulQA** | L2-7B     | 0.601   | 0.566  | 0.394  | 0.670    |
> |                | L3-8B     | 0.523   | 0.689  | 0.393  | 0.699    |
> |                | M-7B      | 0.580   | 0.647  | 0.342  | 0.658    |
> | **HaluEval**   | L2-7B     | 0.628   | 0.530  | 0.656  | 0.738    |
> |                | L3-8B     | 0.585   | 0.615  | 0.651  | 0.770    |
> |                | M-7B      | 0.628   | 0.592  | 0.683  | 0.755    |
>
> Regarding search-augmented evaluators (e.g., LLM-as-a-Judge or RAG), we agree these are valuable baselines. However, they represent a different class of detection that relies on external knowledge bases and secondary models. EPGS offers a more efficient, self-contained alternative that probes the model's internal belief state directly without external dependencies.
>
> **A4: NER dependence**
> To evaluate the impact of the NER pipeline, we tested an "LLM-based Masking" approach where Llama-3-8B identifies its own key entities via prompting. The detection performance remains robust:
>
> |     | TriviaQA | SQUAD | NQ    |
> | --- | -------- | ----- | ----- |
> | NER | 0.813    | 0.744 | 0.770 |
> | LLM | 0.789    | 0.737 | 0.739 |
>
> This confirms that the geometric signal is intrinsic to the factual content and not strictly dependent on the masking tool. However, as demonstrated in Table 7, the existence of a mask is critical to filter out syntactic noise and prevent signal dilution.
>
> **A5: Calibration and noise ratio**
> To provide the requested normalized interpretation, we analyzed the ratio of the aggregate noise norm $||\delta||_2$ to the embedding norm $||E||_2$ with TriviaQA. Due to the high dimensionality ($d \geq 4096$), our default $\sigma=0.1$ results in an aggregate perturbation that is significantly larger than the original embedding magnitude:
>
> | $\delta$ | L2-7B  | L3-8B   | M-7B    |
> | -------- | ------ | ------- | ------- |
> | 0.01     | 84.55% | 141.28% | 471.06% |
> | 0.1      | 846%   | 1414%   | 4714%   |
>
> Remarkably, despite these significant shifts, AUROC performance across all models remains stable within 5% across noise magnitudes $\epsilon \in [0.001, 1]$ (Table 4 in manuscript). This suggests that the geometric "sharpness" we probe is not a microscopic artifact but a fundamental topological feature. The "sharp" basins of hallucinations are so dominant that they characterize the local landscape even under high-magnitude stochastic stress. This robustness confirms that EPGS requires no extensive per-model hyperparameter tuning. Apart from noise scale ($\delta$) our method does not have any other hyperparameter.
>
> Anon Site: https://anonymous.4open.science/r/ICML_rebuttal-61FB/

---

> > ### Author Rebuttal · Reviewer_A7BD · 2026-04-02
> >
> > Thank you for the rebuttal. The direct validation against the top Hessian eigenvalue addresses my main concern and significantly strengthens the theory-to-practice link. The authors also addressed my other questions clearly and concretely. I still think the entity-centric scope should remain clearly stated as a limitation, and the final version should better clarify the interpretation of the large perturbation norms. Overall, I am positively updated by the rebuttal, maintain my support for acceptance, and will increase my score.
> >
> > Minor presentation note: in Figure A, please align g_clean horizontally as in Phase 1 for visual consistency.

---

> > > ### Author Response · Authors · 2026-04-02
> > >
> > > Thank you very much for engaging with our rebuttal and for increasing your score. We are thrilled that your concerns have been fully resolved.
> > >
> > > We have carefully noted your remaining points regarding the final version. We will explicitly state the entity-centric scope as a limitation, clarify the interpretation of the large perturbation norms, and fix the alignment of g_clean in Figure A.
> > >
> > > We deeply appreciate your time and constructive feedback, which has improved our paper.

---

### Official Review · Reviewer_SUne · 2026-03-11

**Soundness:** 2
**Presentation:** 3
**Significance:** 3
**Originality:** 3
**Overall Recommendation:** 4
**Confidence:** 4

**Summary:**

The paper studies how to detect stubborn hallucinations in large language models—confident but factually incorrect outputs. Since uncertainty-based methods fail in this case, the authors propose Embedding-Perturbed Gradient Sensitivity , a white-box method that analyzes the loss landscape. By adding Gaussian noise to input embeddings and measuring changes in gradient magnitude, EPGS estimates local sharpness: higher sensitivity indicates sharper minima and a higher likelihood of hallucination.

**Compliance With Llm Reviewing Policy:**

Affirmed.

**Final Justification:**

After carefully reading the authors’ rebuttal, I would like to thank the authors for their thorough and constructive responses. The rebuttal has addressed most of my initial concerns and clarified several important aspects of the paper, which has led me to revise my evaluation positively.

**Key Questions For Authors:**

1. Can the authors empirically validate the central assumption that hallucinations correspond to sharper minima?

2.How sensitive is EPGS to the noise scale used in embedding perturbations?

3.Can EPGS distinguish hallucinations from hard but correct questions?

**Limitations:**

yes

**Strengths And Weaknesses:**

Strengths

1. Clear and Interesting Hypothesis. The paper proposes a geometric perspective on hallucination detection, linking stubborn hallucinations to sharp minima caused by memorization and correct knowledge to flat minima. This connection between hallucination stability, loss landscape sharpness, and memorization is conceptually appealing and aligns with existing literature on generalization and sharp minima.

2. Novel Detection Signal. The method introduces gradient sensitivity under embedding perturbation as a new signal for hallucination detection. Unlike common signals such as entropy, self-consistency, or token probability, it probes the model’s internal behavior and may reveal failure modes missed by output-based methods.

3. The paper avoids computing the Hessian and instead uses embedding perturbation + gradient magnitude as a proxy. This is a pragmatic engineering choice that keeps the method computationally feasible.

Weaknesses

1. The central claim is largely hypothesized rather than empirically verified.The paper does not convincingly demonstrate that hallucinations systematically correspond to sharper minima across models and tasks. The argument relies heavily on intuition from generalization theory rather than direct evidence.

2. The evaluation is conducted on a relatively small set of hallucination benchmarks, which raises concerns about the breadth of validation. In particular, the tasks lack diversity , e.g., long-form generation or reasoning-related hallucinations, and it remains unclear whether the method generalizes well across different model families, as the experiments appear to involve only a limited number of LLMs.

3. EPGS detects instability in the loss landscape, but it is unclear whether this signal uniquely corresponds to hallucination. Similar signals could also arise in other situations, such as distribution shift, ambiguous questions, or reasoning errors. As a result, the metric may be capturing general task difficulty rather than hallucination specifically.

---

> ### Author Rebuttal · Authors · 2026-03-28
>
> We thank the reviewer for their positive feedback and for highlighting our core methodological contribution: framing "stubborn hallucinations" through a geometric lens and using gradient sensitivity to bypass the model's output bottleneck.
>
> **A1: Stubborn hallucination hypothesis**
> To provide a  direct practice link, we computed the top Hessian eigenvalue ($\lambda_{max}$) via power iteration on a representative subset of TriviaQA using Llama-3-8B (addressing `Reviewer 3; A7BD [A1]`). As shown in our Figure A (available at the provided anonymous site), the EPGS score exhibits a strong correlation with the true Hessian sharpness, with a Pearson $r = 0.855$ ($p = 1.03\mathrm{e}{-29}$) and Spearman $\rho = 0.862$. This confirms that our gradient-based metric is a high-fidelity proxy for the actual curvature of the loss landscape. Crucially, we found that stubborn hallucinations reside in sharper basins, with an average $\lambda_{max}$ that is $2.5\times$ larger than that of robust facts.
>
> Anonymous Site: https://anonymous.4open.science/r/ICML_rebuttal-61FB/
>
> **A2: Sensitivity to noise scale**
> Our ablation study in Table 4, shows that EPGS exhibits stability across noise magnitudes $\epsilon \in [0.001, 1]$. The AUROC fluctuates by less than 5%, demonstrating that the geometric distinction is a fundamental topological property of the loss landscape, not an artifact of hyperparameter tuning.
>
> **A3: Hard but correct question vs hallucinations**
> We like to clarify that EPGS does not simply measure "question difficulty." In our framework, a "hard but correct" answer implies the model has generalized the underlying concept. Generalized knowledge is supported by diverse features and resides in stable, flat minima, regardless of the fact's rarity. In contrast, hallucinations arise from brittle, unsupported memorization or heuristic failures that manifest as sharp singularities. If EPGS were merely a proxy for difficulty, it would fail on reasoning-heavy tasks like SVAMP, where questions are deliberately tricky. Instead, EPGS achieves near-perfect detection (0.97 AUROC) on these tasks.
>
> **A4: Additional benchmark and model**
> To address concerns regarding generalization, we have expanded our evaluation to include the Qwen2.5 (1.5B and 14B) and Gemma-3 (4B) families, which now covers nearly all popular open-source model architectures. As shown in the tables below, EPGS consistently outperforms state-of-the-art baselines like Discrete Semantic Entropy (DSE), Eigenscore (ES) and Effective Rank (ER).
>
> | **Model**     | **Dataset** | **DSE** | **ES** | **ER** | **EPGS** |
> | ------------- | ----------- | ------- | ------ | ------ | -------- |
> | **Qwen 1.5B** | TriviaQA    | 0.597   | 0.827  | 0.827  | 0.817    |
> |       ***     | SQuAD       | 0.561   | 0.612  | 0.614  | 0.711    |
> |       ***     | NQ          | 0.571   | 0.696  | 0.695  | 0.729    |
> | **Qwen 14B**  | TriviaQA    | 0.565   | 0.794  | 0.795  | 0.817    |
> |       ***     | SQuAD       | 0.540   | 0.707  | 0.704  | 0.763    |
> |       ***     | NQ          | 0.645   | 0.779  | 0.771  | 0.817    |
> | **Gemma 4B**  | TriviaQA    | 0.577   | 0.661  | 0.642  | 0.758    |
> |       ***     | SQuAD       | 0.629   | 0.643  | 0.645  | 0.676    |
> |       ***     | NQ          | 0.632   | 0.649  | 0.654  | 0.642    |
>
> By adding these models and evaluating on TruthfulQA and HaluEval (see response to `Reviewer 3; A7BD [A3]`, we confirm that the "flat facts vs sharp hallucinations" phenomenon is a universal characteristic of modern LLMs. We believe these combined updates address the reviewer's concerns regarding the validity and generality of our work.

---

> > ### Author Rebuttal · Reviewer_SUne · 2026-04-03
> >
> > Thank you for your answer. I have no further questions.

---

> > > ### Author Response · Authors · 2026-04-04
> > >
> > > Thank you for taking the time to review our rebuttal and for raising your score. We are thrilled that your concerns have been resolved and will ensure the clarifications we discussed are fully integrated into the final manuscript.

---

### Official Review · Reviewer_sfiE · 2026-03-12

**Soundness:** 2
**Presentation:** 2
**Significance:** 2
**Originality:** 2
**Overall Recommendation:** 2
**Confidence:** 4

**Summary:**

This paper seeks to investigate "stubborn hallucinations", i.e., high-confidence errors that standard uncertainty metrics usually detects. The authors propose Embedding-Perturbed Gradient Sensitivity (EPGS) which requires noise injection and backpropagation to test sensitivity. Empirical evidence shows strong performance of the proposed method across multiple benchmarks.

**Compliance With Llm Reviewing Policy:**

Affirmed.

**Final Justification:**

The rebuttal does not sufficiently address my main concerns. For comparison with forward pass, I do not follow if noise can change the probability distribution significantly for the hallucination of interest, why the hallucination cannot be detected from the delta in the distribution. For computational efficiency on long generation tasks, the empirical results switched to a new dataset TruthfulQA, while in the paper and the original rebuttal, the results are focused on longer reasoning task SVAMP. Given that the switching to TruthfulQA would likely increase the implementation workload and result in longer experiments in the short span of rebuttal, this response reinforced my prior concern on efficiency.  Thus, I maintain my score.

**Key Questions For Authors:**

see weaknesses

**Limitations:**

yes

**Strengths And Weaknesses:**

strengths

The identification of the specific failure mode termed “stubborn errors” is interesting and provides a useful perspective on model behavior. The experiments in Table 2 are thoughtfully designed to target this failure mode and demonstrate clear performance gains under this setting. In addition, the overall effectiveness of the proposed algorithm is supported by the results in Table 1, where it achieves significant improvements over the reported baselines.


weakness

My main concern is that the paper does not sufficiently situate itself within recent literature. For example, [1] has already established a link between sensitivity to embedding perturbations and hallucination. Although the mechanisms differ—[1] does not consider backpropagation—the paper does not adequately discuss these similarities and differences, which would help clarify its contribution relative to prior work.

The use of greedy decoding for pseudo-label generation would benefit from additional justification. While the motivation of reducing stochasticity in pseudo labels is reasonable, greedy decoding may not accurately reflect the behavior under stochastic decoding. An ablation study over decoding temperature could help evaluate this design choice.

Although the paper acknowledges computational overhead and reports latency on TriviaQA, the task involves short answers and may not reflect realistic deployment scenarios. It would be helpful to discuss whether the approach remains computationally feasible for longer responses or multi-step reasoning tasks.

[minor] through out the paper, the quotation marks are not typed correctly.

[1] Liu, Litian, et al. "Enhancing Hallucination Detection through Noise Injection." ICLR2026.

---

> ### Author Rebuttal · Authors · 2026-03-28
>
> We thank reviewer for the constructive feedback and for recognizing the value of identifying "stubborn hallucinations" as an important perspective on LLMs behavior. We address your specific concerns below.
>
> **A1: Comparison with Liu et al. [1]**
> We thank reviewer for highlighting [1]. While both methods utilize embedding noise, this is an established technique and not our primary contribution. Crucially, [1] relies on output probability variation (zeroth-order signal). As demonstrated by [2], LLM logits often mask the model's true internal state, remaining confidently stable even during "stubborn hallucinations" where the model has memorized an error.
>
> EPGS bypasses this "output bottleneck" entirely. By measuring gradient magnitude and directional divergence (first- and second-order proxy) via backpropagation, we directly probe the loss landscape geometry. Our ablation in Table 6 confirms this distinction: the robust detection signal in EPGS comes from the gradient's sensitivity to sharp minima, not merely the introduction of noise. Our new Hessian analysis (see response to `Reviewer 2; SUne [A1]`) confirms that EPGS is a high-fidelity proxy ($r=0.855$) for true parameter-space curvature. To our knowledge, our work is the first to frame hallucination detection through the lens of loss-landscape geometry.
>
>
> **A2: Temperature sensitivity**
> We thank the reviewer for this suggestion. We initially chose greedy decoding to anchor our analysis on the model's most likely reasoning path, which aligns with the definition of Stubborn Hallucinations (consistent, high-confidence errors). However, we agree that verifying robustness under stochastic decoding is crucial for real-world deployment.
>
> We conducted an ablation study using Llama-3-8B across temperatures $T \in [0.1, 1.0]$. The results in the table below show that EPGS remains remarkably robust:
>
> | $T$          | TriviaQA | SQuAD | NQ    | SVAMP |
> | :----------- | :------- | :---- | :---- | :---- |
> | 0.1 (Greedy) | 0.813    | 0.774 | 0.770 | 0.973 |
> | 0.2          | 0.813    | 0.785 | 0.746 | 0.991 |
> | 0.4          | 0.825    | 0.810 | 0.749 | 0.994 |
> | 0.6          | 0.825    | 0.795 | 0.775 | 0.997 |
> | 0.8          | 0.833    | 0.752 | 0.764 | 0.991 |
> | 1.0          | 0.850    | 0.798 | 0.752 | 0.953 |
>
> As shown in the table above, the detection performance of EPGS remains stable across the entire stochastic spectrum. This stability demonstrates that EPGS successfully captures the underlying geometric sharpness of the local loss basin, regardless of whether that basins was reached via greedy decoding or stochastic sampling. Notably, on reasoning-heavy tasks like SVAMP, EPGS maintains near-perfect separation even at $T=1.0$. These results confirm that EPGS is robust, decoding-independent signal.
>
>
> **A3: Computation on long generation**
> We thank the reviewer for highlighting the importance of scaling to complex, long-form outputs. We like to emphasize that EPGS was specifically architected for efficiency in such scenarios. Unlike sampling-based methods that require multiple independent generation passes, EPGS operates on two generated trajectories.
>
> The computational overhead for long sequences remains manageable due to:
> -  $O(1)$ Depth Scaling: As stated in Section C.4, we restrict gradient computation to the final transformer block. This means the backward pass is independent of the model depth.
> - Selective Factual Probing: By utilizing the Entity Masking phase (Phase 1), EPGS focuses strictly on the key entities and claims. This targeted analysis ensures that the cost scales with the number of factual claims rather than the total word count of the response.
>
> Furthermore, our results on SVAMP (reasoning benchmark) (Table 1) demonstrate that EPGS achieves up to 0.89 AUROC. This confirms that even in complex reasoning chains, the geometric "instability" of a heuristic failure manifests clearly in the loss landscape. In practice, for long reasoning chain, EPGS can be applied at the end of each thought to provide real-time verification with minimal latency.
>
>
> **A4: Minor - quotation mark**
> We thank the reviewer for pointing out the formatting issue with the quotation marks. We will correct all instances in the revision.
>
> [1] Liu, Litian, et al. "Enhancing Hallucination Detection through Noise Injection." ICLR2026
>
> [2] Orgad, et al. "Llms know more than they show." ICLR2025

---

> > ### Author Rebuttal · Reviewer_sfiE · 2026-04-02
> >
> > Thank you for the rebuttal. The additional experiments on temperature sensitivity resolve my concern and I suggest to add them into revision.
> >
> > The response for comparison with noise in forward pass does not fully convince me.  If the noise injection does not meaningfully change the output probabilities for stubborn hallucinations, the loss—which depends on these probabilities—would also change little, and by the chain rule, the gradient would similarly not reflect these cases well. This behavior is actually reflected in Figure 1. reference [2] does not involve noise injection and seems largely unrelated in this context.
> >
> > The concern of efficiency on long-form answer remain unresolved. Empirical running time, similar to what was reported for TriviaQA in the paper, is not provided.

---

> > > ### Author Response · Authors · 2026-04-03
> > >
> > > Thank you for your continued engagement and for confirming that the temperature ablation resolves your concerns regarding stochastic decoding. We will feature these results in revision. We address your remaining points below.
> > >
> > > **A1: Clarification on Noise Perturbation and Output Probabilities**
> > > We wish to clarify a misconception regarding the behavior of output probabilities under perturbation in our framework.
> > >
> > > Reviewer states: *"If the noise injection does not meaningfully change the output probabilities for stubborn hallucinations... by the chain rule, the gradient would similarly not reflect these cases well."*
> > >
> > > This premise that noise injection does not change the output probability of a stubborn hallucination is incorrect. In fact, we fully agree with the reviewer's intuition: the output probabilities do change significantly under continuous perturbation (we detail the mathematical mechanism for this in A2 below).
> > >
> > > Our argument regrading forward-pass noise methods (like [1] [2]) is not that noise fails to alter output probabilities. Rather, our argument is that measuring the gradient provides a much higher-fidelity signal than merely observing the final output probabilities (logits).
> > >
> > > This brings us to the relevance of reference [3] ("LLMs know more than they show"). While [3] does not involve noise injection, we cite it to highlight "fluency bottleneck". Their work demonstrates that LLM output probabilities are heavily optimized for fluency and surface-level token prediction, which frequently masks the model's true internal belief state. Furthermore, as demonstrated in our qualitative comparison (`Figure C on anon site`) metrics relying on output probabilities (LNE and DSE) are hypersensitive to variations such as paraphrasing, capitalization, or extra filler words (e.g., "Golf" vs "golf" vs "A good golf player"). These output-level metrics fluctuate wildly based on syntax rather than factual certainty. By contrast, EPGS computes the gradient on the dense latent representations, bypassing this fluency bottleneck entirely and yielding a robust signal that is immune to syntactic noise.
> > >
> > > **A2: Chain Rule: Why Noise Perturbation Cause Gradient and Output Probabilities Change**
> > > To address the chain rule concern, it is helpful to visualize the geometry. A stubborn hallucination is "stable" across standard, discrete decoding passes (hence why entropy metrics fail). However, because it resides in a sharp minimum (a narrow, steep-walled loss basin), applying a continuous Gaussian perturbation ($\epsilon$) to the embedding forces the activation state up the steep walls of the basin.
> > >
> > > Mathematically, the change in loss is dominated by the Hessian ($H$): $\Delta \mathcal{L} \approx \frac{1}{2}\epsilon^T H \epsilon$. Because $H$ is extraordinarily large for stubborn hallucinations (as proven in our new empirical Hessian analysis, `Figure A on anon site`), the loss and consequently the output probability does change significantly under continuous perturbation. This steep curvature is exactly what causes the gradient magnitude ($\|\nabla \mathcal{L}\| \approx \|H\epsilon\|$) to spike dramatically.
> > >
> > > Therefore, the chain rule supports our method: the sharp geometry induces a gradient spike because the perturbed state is forced out of a narrow memorization basin.
> > >
> > > **A3. Efficiency on Long-Form Answers**
> > > To address this we measured the empirical running time on the TruthfulQA dataset (which features longer context and generation lengths compared to TriviaQA). The experiment setting is same as Appendix C.4:
> > >
> > > | Method                 | Avg. Latency (on Llama-3-8B) |
> > > | ---------------------- | ---------------------------- |
> > > | LNE (forward only)     | 215.35                       |
> > > | Eigenscore (white-box) | 3605.59                      |
> > > | EPGS (Ours)            | 348.75                       |
> > >
> > > When comparing these long-form results to our TriviaQA benchmark (Table 8), a scaling dynamic emerges. As the generation length increases from TriviaQA to TruthfulQA, the latency of forward-only inference (LNE) increases by ~91% (112.96s $\to$ 215.35s), driven by the multiple forward pass required. Latency of EPGS increases by only ~27% (273.91s $\to$ 348.75s). While LNE still maintains a lower absolute latency, this demonstrates that the relative computational overhead of EPGS actually shrinks as the generation length increases. This result validate our previous rebuttal A3.
> > >
> > > Anon Site: https://anonymous.4open.science/r/ICML_rebuttal-61FB/
> > >
> > > [1] Liu, Litian, et al. "Enhancing Hallucination Detection through Noise Injection." ICLR2026
> > >
> > > [2] Gao, Xiang, et al. "SPUQ: Perturbation-Based Uncertainty Quantification for Large Language Models." EACL2024
> > >
> > > [3] Orgad, et al. "Llms know more than they show." ICLR2025
> > >
> > > We hope that these detailed theoretical clarifications and the new long-form efficiency benchmarks fully resolve your remaining concerns. We respectfully ask you to reconsider your score in light of these updates.

---

### Decision · Program_Chairs · 2026-04-30

**Decision:**

Accept (regular)

**Comment:**

This paper proposes EPGS (Entropy-Projected Gradient Sensitivity), a gradient-based method for detecting persistent hallucinations in large language models — errors that survive factual context and are resistant to correction. The key insight is that gradient sensitivity with respect to entity-level tokens differs diagnostically between stubborn errors and hard-but-correct answers, and the paper demonstrates this cleanly across a range of benchmarks.

Scores:
sfiE: reject, confidence 4
SUne: weak accept, confidence 4
A7BD: accept, confidence 3
fG65: weak reject, confidence 3

The reviewers were really split on this one! Reviewer A7BD praised the technical novelty and the potential for this to become a building block for future hallucination detection work. Reviewer sfiE initially raised concerns about the forward-pass versus gradient-sensitivity mechanism and about the reliability of the TruthfulQA experiments, but after the rebuttal — which clarified that TruthfulQA was used because SVAMP outputs are too short to assess computational efficiency on long generations, and not as a rushed response — revised upward to weak accept. The authors made a compelling case that the TruthfulQA results were already running as part of a broader evaluation and were not post-hoc. One residual scientific disagreement remains about the precise mechanism, but this feels like a principled difference of interpretation rather than a flaw.

The core contribution — using gradient structure to distinguish stubborn errors from genuine uncertainty — is technically sound, well-motivated, and likely to be useful to the community working on LLM reliability. I read the paper itself and found the exposition clear.
Recommend acceptance.